# Conservation Tillage and Weed Management Influencing Weed Dynamics, Crop Performance, Soil Properties, and Profitability in a Rice–Wheat–Greengram System in the Eastern Indo-Gangetic Plain

Bushra Ahmed Alhammad [1], Dhirendra Kumar Roy [2,*], Shivani Ranjan [2], Smruti Ranjan Padhan [3], Sumit Sow [2], Dibyajyoti Nath [4], Mahmoud F. Seleiman [5,6,*] and Harun Gitari [7]

1 Biology Department, College of Science and Humanity Studies, Prince Sattam Bin Abdulaziz University, Al Kharj Box 292, Riyadh 11942, Saudi Arabia
2 Department of Agronomy, Dr. Rajendra Prasad Central Agricultural University, Pusa, Samastipur 848125, India
3 Division of Agronomy, ICAR-Indian Agricultural Research Institute, Pusa Campus, New Delhi 110012, India
4 Department of Soil Science, Dr. Rajendra Prasad Central Agricultural University, Pusa, Samastipur 848125, India
5 Plant Production, College of Food and Agriculture Sciences, King Saud University, Riyadh 11451, Saudi Arabia
6 Department of Crop Sciences, Faculty of Agriculture, Menoufia University, Shibin El-Kom 32514, Egypt
7 Department of Agricultural Science and Technology, School of Agriculture and Environmental Sciences, Kenyatta University, Nairobi P.O. Box 43844-00100, Kenya
* Correspondence: dr_dhirendra_krroy@yahoo.com (D.K.R.); mseleiman@ksu.edu.sa (M.F.S.)

**Abstract:** A three-year field experiment was carried out to assess the efficacy of various tillage and residue management practices, as well as weed management approaches, in a rice–wheat–green gram rotation. The treatments included: conventional till transplanted rice–conventional till wheat–fallow ($T_1$); conventional till transplanted rice–zero-till wheat–zero-till green gram ($T_2$); conventional till direct-seeded rice—conventional-till wheat—zero-till green gram ($T_3$); zero-till direct-seeded rice—zero-till wheat—zero-till green gram ($T_4$); zero-till direct-seeded rice + residue zero-till wheat + residue zero-till green gram ($T_5$). In weed management, three treatments are as follows: recommended herbicides ($W_1$); integrated weed management ($W_2$); and unweeded ($W_3$). The integrated weed management treatment had the lowest weed biomass, which was 44.3, 45.3, and 33.7% lower than the treatment $W_3$ at 30 and 60 days after sowing and harvest, respectively. $T_1$ grain and straw yielded more than $T_2$ in the early years than in subsequent years. The conventional till transplanted rice–zero-till wheat–zero-till green gram system produced 33.6, 37.6, and 27.7% greater net returns than the zero-till direct-seeded rice—zero-till wheat—zero-till greengram system, respectively. Conventional till transplanted rice–conventional till wheat–fallow had the biggest reduction (0.41%) in soil organic carbon from the initial value. The findings of the study demonstrated that adopting the transplanting method for rice, followed by zero tillage for wheat and green gram, enhanced productivity and profitability, while simultaneously preserving soil health.

**Keywords:** productivity; profitability; rice–wheat–green gram; soil health; tillage

## 1. Introduction

The rice–wheat cropping system, which spans roughly 14 million hectares in South Asia's Indo-Gangetic Plain (IGP) [1], has several obstacles that have hampered its efficacy. South Asia has been dealing with difficulties such as diminishing soil health [2], groundwater depletion [3], growing climatic variability [4], air pollution because of residue burning [5], and shifting socioeconomic conditions. These difficulties have had a substantial influence on the region's rice–wheat farming systems.

The Eastern Indo-Gangetic Plain (EIGP) is primarily made up of marginal farmers, who account for roughly 90% of the population, with a per capita income per household per year of INR 62,631 (USD 835), which is significantly lower compared to the county's average of INR 94,130 (USD 1256) [6]. As the population in Eastern India grows, there is an urgent need to raise agricultural intensity to satisfy the region's food and nutritional needs [7]. Initiatives have been launched to kickstart a second Green Revolution in Eastern India to achieve food security. However, the rice–wheat cropping system (RWCS) on the EIGP has many difficulties. Long-duration paddy types dominate most rice fields in the region, resulting in late transplanting and delayed harvesting. As a result of the delayed planting of wheat in the rice–wheat cropping system, yields have been lowered and grain quality has been affected due to heat stress during the grain-filling stage [8–10]. Furthermore, the management of rice residues, which are frequently left loose and scattered after harvest, is a substantial impediment since they interfere with tillage operations and the sowing of the next wheat crop [11]. Due to its cost-effectiveness, burning the residual rice and wheat residues is a widespread practice among local farmers in the EIGP region [12]. However, this burning process results in substantial nutrient losses, including 5.5 kg of N, 2.3 kg of P, 25 kg of K, and 1.2 kg of S, as well as organic carbon [13]. Furthermore, according to Jain et al. [4], agricultural residue burning adds to air pollution with emissions of 8.77 Mt CO, 0.23 Mt NO, 141.15 Mt $CO_2$, and 0.12 Mt $NH_3$. A considerable fraction of the nutrients found in crop residues is lost by gas and particle emissions, including 25 percent phosphorus, 80 to 90 percent nitrogen, 50 percent sulfur, and 20 percent potassium. These emissions, coupled with carbonaceous matter, considerably contribute to air pollution and global warming [14]. Greenhouse gas emissions (6266 Gg per year) can be lowered and soil health enhanced by minimizing agricultural residue burning and integrating residues into the soil [15]. As a result, an alternative production system that addresses these challenges by lowering production costs, conserving natural resources, reducing labor and time requirements, effectively controlling weeds, increasing productivity, and protecting the environment is urgently needed [16].

Conservation tillage is being adopted by an increasing number of farmers in South Asia as an alternate way to address rising difficulties. CA entails reducing or eliminating soil disturbance and keeping agricultural leftovers on the soil surface [17]. This transition towards CA technologies not only enhances production and revenue, but also addresses many challenges, such as restricted land size, diminishing agricultural output, growing cultivation expenses, farming risk, and the issue of climate change [18]. These difficulties represent serious concerns for livelihood security, especially for small-scale farmers. Zero tillage with residue retention has yielded excellent results, including a 5.8% increase in yields, a 25.9% increase in net income, and a 12.33% decrease in global warming potential [18,19]. While large-scale mechanized farms in the Americas and Australia have effectively embraced CA systems [20,21], smallholder farmers' adoption has been slower [22]. Furthermore, different regions' CA practices and cropping systems differ from those in the Eastern Indo-Gangetic Plain (EIGP). Due to the limited availability of resources such as sowing tools and pesticides, as well as traditional agricultural ideas, the broad adoption of zero tillage (ZT) in the EIGP has been hampered [23,24]. Although some CA features, such as ZT and residue retention, have been largely implemented in various crops, there is still a long way to go [25]. Various problems must be solved to encourage the broad adoption of CA systems within farming communities, with specific concerns varying based on the local situation.

Soil microorganisms are important in sustaining soil ecosystem health because they regulate several biochemical cycles and contribute to overall soil quality. Conservation agricultural practices have a large influence on soil microbial populations, making them useful markers of soil health and ecosystem resilience. Phosphate-solubilizing microorganisms (PSMs) are particularly essential among these beneficial microbial groups because they have the potential to hydrolyze phosphorus, boosting plant nutrition and enriching the soil [26,27]. However, both the physicochemical soil properties and agronomic practices

impact the availability and activity of PSMs [28]. Factors such as soil pH, poor soil structure, low levels of soil organic carbon (SOC), and fluctuations in nitrogen, phosphorus, and potassium availability can limit the presence and functioning of PSMs [29,30]. In contrast to traditional tillage, zero tillage with residue retention reduces soil disturbance, which helps soil microbial populations in a variety of ways. It encourages microbial variety, increases SOC accumulation, preserves fungal hyphae, maintains soil food webs, and creates specialized microsites and microbial niches, all of which stimulate microbial activity and proliferation [29,31]. Conservation agricultural practices generate favorable circumstances for microbial communities to flourish by protecting the integrity of the soil structure and retaining organic wastes on the soil surface.

The successful implementation of conservation tillage in the rice (*Oryza sativa*)–wheat (*Triticum aestivum*)–green gram (*Vigna radiata*) cropping system has become dependent on the deployment of integrated weed management [32]. Tillage procedures that uproot, disturb, and bury weeds deep in the soil, limiting their emergence, are used in conventional tillage practices to accomplish effective weed management [14]. Weed seeds, on the other hand, tend to collect on the soil surface under conservation tillage systems that minimize or eliminate tillage, leading to increased weed development. In California, lower tillage intensity and frequency correlate to increased weed infestation levels. Furthermore, shifting from conventional to conservation-based farming might cause a shift in the composition of weed species within the crop field [33]. Furthermore, crop residues on the soil surface might intercept and bind herbicides, decreasing their ability to reach the soil surface. Consequently, the use of post-emergence herbicides in conservation tillage has become critical for weed control. Zero tillage also avoids bringing weed seed back up from the subsoil, residue cover impedes weed growth, and crop rotation reduces weeds, so conservation tillage, if well managed, can reduce weed problems in the medium to long term [34]. Furthermore, weed control practices have been widely encouraged across various tillage systems to boost soil fertility and crop production [35].

The majority of research in the Eastern Indo-Gangetic Plain (EIGP) has focused on zero-tillage practices for specific crops under the rice–wheat cropping system (RWCS). Thus, the major goal of this study was to assess the best tillage and weed control practices for increasing rice–wheat–green gram system productivity, soil fertility, and profitability. We hypothesized that the adoption of conservation tillage and weed management practices would result in improved crop yields, enhanced soil chemical and microbial properties, increased net income, and reduced weed infestation in the field.

## 2. Materials and Methods

### 2.1. Study Site

The experiment was conducted in an agricultural research center in Pusa, Bihar, India, with precise coordinates of 85° 48′ E longitude, 25° 59′ N latitude, and 52.92 m above mean sea level (Figure 1). This study was carried out between 2013 and 2016 as part of the Project Directorate on Weed Research's research program in Jabalpur. The trial lasted 36 months, with rice (*Oryza sativa* L.) grown during the rainy season (July–November), wheat (*Triticum aestivum* L.) grown from November to April, and green gram (*Vigna radiata* (L.) Wilczek) grown from April to July during the dry season. The climate in the study region is subtropical hot and humid, with a mean annual rainfall of 1210 mm. The majority of the rainfall, 75–80%, falls between July and September. The coldest temperature in January is around 5 °C, while the highest temperature in May is around 40.5 °C. During the research period, there were two instances of high rainfall: 255 mm in August 2014 and 316.2 mm in August 2015. The wet rice crop was lodged as a result of the severe rains in 2014. Figure 2 depicts the average weekly temperature, relative humidity, and monthly rainfall data obtained throughout the research from 2013–2014 to 2015–2016.

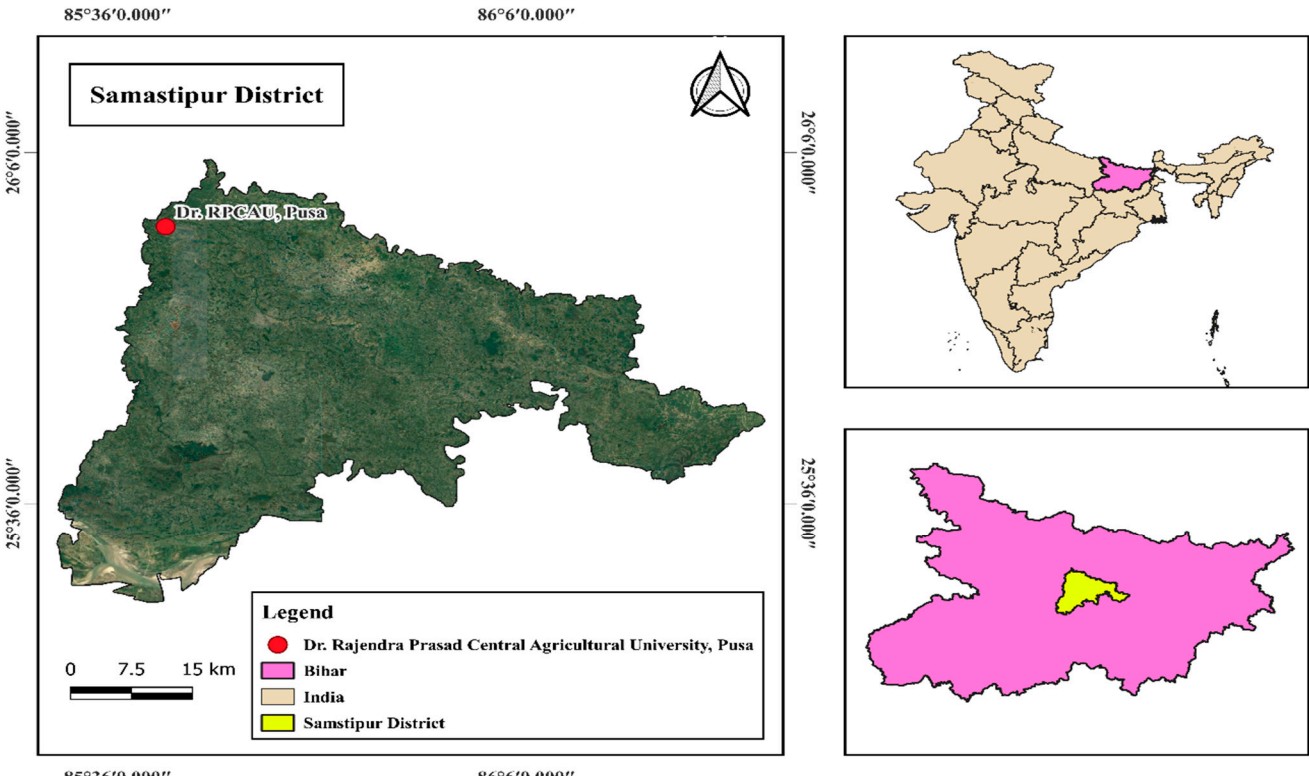

**Figure 1.** The geographical location map of the study area.

### 2.2. Weather Details

During the three years of the experiment (2013–2014 to 2015–2016), there were significant fluctuations in rainfall patterns, both in amount and distribution, raising concerns about rainfall uncertainty in the Eastern Indo-Gangetic Plain (EIGP). The mean maximum and minimum temperature, relative humidity, and rainfall received during the crop period are shown in Figure 2. Rainfall received from June to September accounted for 80–93% of the total yearly rainfall measured throughout the research period. During the rice season (July–November), the greatest reported rainfall was 811.7 mm in 2014, 160 mm in 2015, and 768.7 mm in 2016 (Figure 2). For the wheat season (November to April), the highest rainfall was 125.61 mm in 2013, followed by 44 mm in 2014, and 10.2 mm in 2015. In 2013–2014, 2014–2015, and 2015–2016, the average morning relative humidity was 81.48%, 87.32%, and 84.64%, respectively. During the same years, the average relative humidity in the evening was 50.42%, 55.36%, and 48% (Figure 2). Similarly, the mean weekly maximum and minimum temperatures for the summer green gram growth season varied between 36.8–40.7 °C and 21.1–23.2 °C, respectively, in all years. During the research years, the total rainfall was 157.2 mm (2014), 256 mm (2015), and 237.9 mm (2016).

### 2.3. Experimental Design and Treatment Details

The study employed a strip plot design, where each plot measured 20 m × 10 m. It consisted of a total of five tillage treatments in the main plots and three weed management treatments in subplots, replicated three times. The detailed treatment combinations are presented in Table 1.

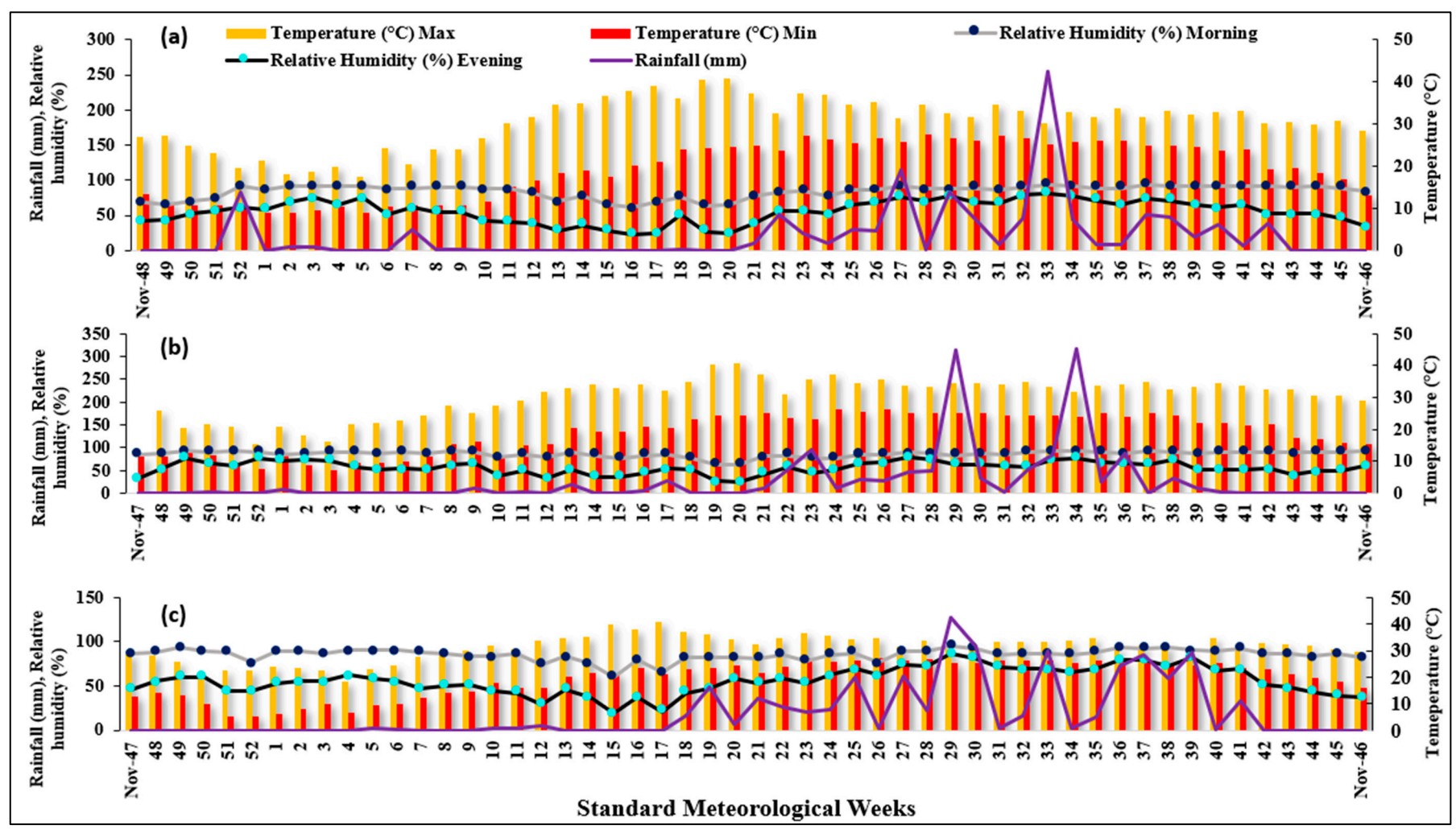

**Figure 2.** The mean weekly maximum temperature, minimum temperature, relative humidity in the morning, relative humidity in the evening, and rainfall for 2013–2014 (**a**), 2014–2015 (**b**), and 2015–2016 (**c**).

**Table 1.** Treatment details of the rice–wheat–green gram system.

| Treatment | Rice | Wheat | Green Gram |
|---|---|---|---|
| Tillage and residue management | | | |
| $T_1$ | CT (T) | CT | Fallow |
| $T_2$ | CT (T) | ZT | ZT |
| $T_3$ | CT (DS) | CT | ZT |
| $T_4$ | ZT (DS) | ZT | ZT |
| $T_5$ | ZT (DS) + R | ZT + R | ZT |
| Weed management | | | |
| $W_1$ | Recommended herbicides | | |
| $W_2$ | Integrated weed management (IWM) | | |
| $W_3$ | Unweeded | | |

CT: Conventional tillage; ZT: Zero tillage; T: Transplanted; DS: Direct-seeded, R-Residue.

*2.4. Crop Management*

2.4.1. Rice

In the experiment, a medium-duration rice variety 'Rajendra Sweta (RAU710-99-22)' was used. For the direct-seeded rice (DSR) treatments, the rice seeds were sown using a Zero-Till Happy Seeder equipped with an inclined-plate seed metering system manufactured by Dasmesh, located in Malerkotla, Sangrur, Punjab, India. During the first to second week of July each year, sowing was performed in rows 22.5 cm apart. For all DSR treatments, the seed rate was set at 50 kg ha$^{-1}$, and a constant seeding depth of 3–4 cm was maintained using the seeder's depth control system. Nurseries were established in conventional transplanted rice (CTR) on the same day as DSR sowing, using the suggested package of practices outlined by Singh et al. [36]. The nurseries were established with a seed rate of 25 kg ha$^{-1}$. After 25 days, the resulting 2–3-week-old seedlings were manually transplanted into puddled fields. The transplantation process for CTR involved placing the seedlings at a spacing of $20 \times 15$ cm, with 2–3 seedlings per hill.

Rice was fertilized with a combination of urea and diammonium phosphate (DAP) at a rate of 120 kg of nitrogen (N) per hectare, along with 60 kg of phosphorous (P$_2$O$_5$) as DAP and 60 kg of potassium (K$_2$O) as muriate of potash (MOP), as described by Jat et al. [37]. During the sowing or transplanting stage, the entire amount of phosphorus and potassium fertilizers, as well as half of the recommended dose of nitrogen, were applied. The base dose of fertilizer was administered to transplanted rice during the final puddling phase, right before seedling transplantation. These fertilizers were drilled into the soil during the planting of direct-seeded rice (DSR) using the Zero-Till Happy Seeder. The remaining two-thirds of the nitrogen was supplied in two equal parts during the crop's mid-tillering and panicle initiation stages, as recommended by Kumar et al. [38]. To manage weeds, pre-sowing applications of pendimethalin at a rate of 1.0 kg a.i. ha$^{-1}$ were applied in all the plots except the unweeded treatments. As per the recommended dose of herbicide treatments, pretilachlor was administered at a dosage of 0.75 kg a.i. ha$^{-1}$ in the prescribed herbicide treatments at 20–25 days after sowing (DAS) in the CT/ZT/ZT + R-DSR treatments and 2–3 days after transplanting (DAT) in the CTR treatment. In the integrated weed management (IWM) treatments, two hand weeding sessions were performed, one at 40–45 DAS and another at 20–25 DAT, besides the pre-sowing application of pendimethalin. The Khurpi (trowel) was commonly used as a tool for weeding in the field.

Each herbicide was dissolved in 500 milliliters of water to make unique stock solutions for the herbicide solutions. Following that, the stock solutions were diluted with water to achieve a spray volume of 500 L per acre. A knapsack sprayer outfitted with a flat fan nozzle was used to apply the spray solution.

Weed density was assessed by counting weeds at three separate times: 30 days after sowing or transplanting (DAS/T), 60 DAS/T, and harvest. A 0.5 m $\times$ 0.5 m quadrate was set in each plot, except for the two boundary rows, and the number of weeds within the quadrate was counted. Weed biomass was measured by physically removing weeds from

the sampling rows above ground with a sickle. The weeds were then sun-dried before being dried in a hot air oven at 60 °C until they reached a consistent dry weight.

To provide sufficient soil moisture for germination, direct-seeded rice (DSR) plots, including the ZT/ZT + R/CT treatments, were sown following the commencement of premonsoon showers. The land was prepped for transplanted rice by rotavator puddling with roughly 7 cm of water. Following the floodwater's receding, further irrigations were administered, and the irrigation schedule accounted for any rainfall occurrences. Depending on the amount and distribution of rainfall, the number of post-sowing irrigations ranged from 3 to 4. Various metrics were measured at regular intervals to determine rice growth and production. Plant height (cm) and effective tiller count (number per square meter) were measured at 30 and 60 DAS/T, as well as at harvest. Plant height was determined by randomly picking ten panicles from each plot. The number of effective tillers was obtained by counting them inside a 1 $m^2$ quadrate from four distinct sites within each plot and taking the average.

To assess the yield, a 10 $m^2$ area was allocated in the center of each plot for harvesting and measuring the grain and straw yields. In most treatments, the crop was picked manually with a sickle, around 15 cm above ground level. The crop was manually picked at a height of roughly 30 cm above ground level in the ZT + R treatments. The grain was sun-dried and manually threshed after harvest. To guarantee precise results, the grain yield was adjusted to a moisture content of 14%.

### 2.4.2. Wheat

Following the rice harvest, cultivation of the popular wheat variety 'HD 2967' was carried out. Wheat was planted in the fourth week of November. Traditional tillage practices, as reported by Singh et al. [36], were used in conventional tillage (CT) plots. In contrast, the zero-till (ZT) and zero-till with residue (ZT + R) plots were seeded directly onto the rice crop residue without any tillage, as shown in Table 1. The wheat plots, whether CT or ZT/ZT + R, were drilled with 22.5 cm rows using a Happy Seeder outfitted with an inclined-plate seed metering system. In all treatments, a seed rate of 100 kg/ha was used, and the seeds were uniformly planted at a depth of around 5 cm. Following the directions provided by Jat et al. [39], the wheat crop obtained 120 kg of nitrogen (N) in the form of urea and diammonium phosphate (DAP), 60 kg of phosphorous pentoxide ($P_2O_5$) as DAP, and 60 kg of potassium oxide ($K_2O$) as muriate of potash (MOP).

Using the Zero-Till Happy Seeder, the full amount of phosphorus and potash, as well as half of the recommended dose of nitrogen, were administered during wheat crop planting. The remaining two-thirds of nitrogen was applied in two equal treatments during the crop's crown root initiation (CRI) and maximum tillering stages. Pre-sowing irrigation was used to establish the wheat crop, followed by four further irrigations at important growth stages: CRI, tillering, blooming, and grain filling. Each irrigation required the application of around 5 $cm^3$ of water. Except for the unweeded treatments, weed control methods were applied, including the common spraying of glyphosate at a rate of 1.0 kg a.i. ha$^{-1}$ before planting. A ready-mix solution of sulfosulfuron (75% WG) + metsulfuron-methyl (5% WG) at a combined rate of 32 (30 + 2) g a.i. ha$^{-1}$ was treated 25 days after sowing (DAS) in the recommended herbicide treatments. In the integrated weed management (IWM) treatments, fenoxaprop ethyl 100 g a.i. ha$^{-1}$ was applied at 25 DAS, and one manual weeding session was conducted at 40–45 DAS.

Several growth and yield characteristics of wheat were recorded at the maturity stage, including the number of effective tillers per square meter, grains per earhead, and test weight (1000-grain weight) at 12% moisture content. Ten spikes were randomly picked from each plot to assess plant height, number of earheads per square meter, number of grains per earhead, and test weight (in grams). Every year, the wheat harvest takes place in the latter week of April. A specified area of 5 m × 2 m (10 $m^2$) positioned in the center of each plot was harvested to measure grain and straw yield. Manual harvesting using a sickle was carried out in all treatments, with the plants being cut approximately 15 cm above ground

level. Subsequently, the harvested wheat underwent manual threshing after appropriate sun drying. The recorded grain yield was adjusted to a moisture content of 12%.

### 2.4.3. Green Gram

The cultivation of a short-duration green gram cultivar, 'SML 668', developed by Punjab Agricultural University in Ludhiana, Punjab, India, was carried out during the study. Following the harvest of wheat, pre-sowing irrigation was applied to all plots, and green gram seeds were sown without any tillage into the wheat crop residue during the final week of April. The sowing process involved using a Zero-till Happy Seeder equipped with an inclined-plate seed metering system, with rows spaced 22.5 cm apart. A seed rate of 20 kg per hectare was utilized, and the seeds were sown at a depth of approximately 5 cm in all treatments, with the seeder's depth control system ensuring uniformity. In all the plots, the pre-sowing application of paraquat herbicide was applied to manage existing weeds except for unweeded treatments. Furthermore, a pre-emergence herbicide, pendimethalin, was applied at a rate of 1 kg a.i. per hectare one day after seeding (except in the unweeded treatments) to control subsequent weed emergence. In the integrated weed management (IWM) treatments, an additional hand weeding session was conducted 20–25 days after sowing (DAS), besides the application of pendimethalin one day after seeding to further control weed growth.

The green gram crop received a fertilization treatment consisting of 18 kg of nitrogen (N) and 46 kg of phosphorus pentoxide ($P_2O_5$) per hectare, which was supplied through diammonium phosphate (DAP) fertilizer. The entire dose of nitrogen and phosphorus was applied at the time of sowing using the Happy Seeder. Two additional irrigations were provided to all plots, with one at 25 days after sowing (DAS) and another at 45 DAS. To determine the grain yield, a designated area measuring 5 m $\times$ 2 m (10 m$^2$) located at the center of each plot was harvested. The recorded grain yield was reported at a moisture content of 12%. Harvesting of the mature pods was undertaken manually using a sickle, ensuring a consistent cutting height of approximately 15 cm above ground level in all treatments, except for the ZT + R treatments, where the crop was manually harvested at a cutting height of approximately 30 cm above ground level.

For the study of weed seed banks, soil samples were collected before the sowing/transplanting of the crop and were consistently irrigated. The number of germinated weed seedlings was regularly recorded at three intervals: 15 days (first flush), 25 days (second flush), and 40 days (third flush).

### 2.5. Soil Sampling and Analysis

Table 2 presents the initial physicochemical properties of the soil at the study site in 2013, before wheat sowing. After the experiment in 2016, soil samples were collected from all plots at a depth of 0–15 cm following the harvest of rice. To ensure representative soil samples, "V"-shaped slices were created, and five random samples were collected. These samples were thoroughly mixed, and approximately 500 g of soil was taken for analysis of the parameters listed in Table 3.

### 2.6. Economic Analysis

An economic analysis was conducted to assess the rice–wheat–green gram cropping system during the cropping year. The analysis involved calculating various economic indicators using the prevailing market prices of inputs and outputs. The cost of cultivation was determined by considering variable costs, excluding land rent. This included expenses for seeds, fertilizers, pesticides, human labor, machinery, irrigation, and other relevant activities. Fixed costs were not taken into account in the analysis.

The labor cost associated with various field activities was assessed based on person-days per hectare, with 8 h being comparable to 1 person-day under Indian labor regulations. The labor cost was estimated by multiplying the labor used in all processes by the government-mandated minimum pay rate stated in the Minimum Pay Act of 1948.

**Table 2.** Initial physicochemical properties of the soil at the study site.

| Parameter | Value | Method Used | Reference |
|---|---|---|---|
| Sand (%) | 24.72 | | |
| Silt (%) | 48.85 | Bouyoucos hydrometer | Piper [40] |
| Clay (%) | 25.77 | | |
| Texture | Silty clay loam | Textural diagram | Black [41] |
| Bulk density (g cm$^{-3}$) | 1.43 | Core sampler | Black [41] |
| Soil pH (1:2.5 soil water suspension) | 8.04 | Potentiometric | Jackson [42] |
| Electrical conductivity (dS m$^{-1}$) | 0.48 | Potentiometric | Jackson [42] |
| Organic carbon (%) | 0.51 | Walkley and Black's rapid titration method | Jackson [42] |
| Available nitrogen (N) (kg ha$^{-1}$) | 247.64 | Alkaline $KMnO_4$ | Subbiah and Asija [43] |
| Available phosphorus ($P_2O_5$) (kg ha$^{-1}$) | 36.38 | Olsen's method | Olsen et al. [44] |
| Available potash ($K_2O$) (kg ha$^{-1}$) | 249.48 | 1 N neutral ammonium acetate method | Jackson [42] |

**Table 3.** Methods used in soil sample analysis.

| Parameter | Method Used | Reference |
|---|---|---|
| Soil pH (1:2.5 soil water suspension) | Potentiometric | Jackson [42] |
| Organic carbon (%) | Walkley and Black's rapid titration method | Jackson [42] |
| Available nitrogen | Alkaline $KMnO_4$ | Subbiah and Asija [43] |
| Available phosphate | Olsen's method | Olsen et al. [44] |
| Available potash | 0.01 N neutral ammonium acetate method | Jackson [42] |
| Azotobacter ($10^4$ cfu g$^{-1}$ soil) | | |
| Total Pseudomonas ($10^5$ cfu g$^{-1}$ soil) | | |
| Total PSB ($10^5$ cfu g$^{-1}$ soil) | | |
| % of P solubilized by Pseudomonas | - | Schmidt and Coldwell [45] |
| Bacillus ($10^5$ cfu g$^{-1}$ soil) | | |
| % of P solubilized by Bacillus | | |
| $CO_2$ evolution (mg kg$^{-1}$) | | Zibilske [46] |

Gross returns (GR) were determined by multiplying the grain yield of each crop (in tons per hectare) by the minimum support price (MSP) offered by the Government of India for the respective years of 2013–2014, 2014–2015, and 2015–2016. The value of the straw was calculated using prevailing local market rates.

Net returns (NR) were calculated as the difference between gross returns and the cost of cultivation (CC) (NR = GR − CC). The benefit-to-cost ratio (B:C ratio) was computed by dividing the gross returns by the cost of cultivation (B:C ratio = GR/CC).

For the economic analysis, exchange rates of 1 USD = INR 60.99 (in 2014), INR 64.13 (in 2015), and INR 67.18 (in 2016) were considered, based on the average exchange rate for the period of 2014–2016 (source: https://www.exchangerates.org.uk (accessed on 12 April 2023)).

### 2.7. Statistical Analysis

Before doing statistical analysis, weed population data were square root transformed ((x + 1)). The analysis was carried out using the CPCS-1 statistical program developed by Punjab Agricultural University, Ludhiana [47]. Analysis of variance (ANOVA) was carried out with a strip plot design on all data relating to weed dynamics, growth, yield characteristics, yield, and economics using the Statistix 8.1 statistical tool (Analytical Software, Tallahassee, FL, USA) [48]. The significance of the treatment effect was determined using an F-test at a 5% level of significance. The least significant difference (LSD) or critical difference (CD) approach was used to assess differences between treatment means [49].

## 3. Results

### 3.1. Weed Dynamics

#### 3.1.1. Weed Seed Bank and Its Dynamics in Soil

The major weed species that emerged from the soil during the three flushes in rice during the grow-out tests conducted on soil samples from the permanent tillage trial before the wet season of 2016 (after three years of completing the trial) were *Echinochloa crusgalli*, *Leptochloa chinensis*, *Cyperus difformis*, *Ammania baccifera*, and *Dactyloctenium aegyptium*. In the first flush, second flush, and third flush, the number of weed seeds that emerged was higher in the ZT(DS)–ZT–ZT (rice–wheat–green gram) treatment compared to all other tillage treatments. The unweeded treatment exhibited the highest weed density, while the IWM ($W_2$) treatment recorded a lower weed density compared to the RDH ($W_1$) treatment in all three flushes (Figure 3a).

A grow-out test was undertaken on soil samples obtained from different soil depths under various treatments before wheat sowing during the dry season of 2015 (after three years of the study). The findings revealed that Phalaris minor, C. album, and M. indica were the most common weed species across all treatments. The ZT(DS)–ZT–ZT treatment had the greatest weed density, followed by ZT(DS) + R–ZT + R–ZT, CT(DS)–CT–ZT, CT(T)–ZT–ZT, and CT(T)–CT–fallow treatments. The weed density was uniform over all three flushes (Figure 3b). Likewise, the unweeded treatment had the highest weed density.

Another grow-out test was undertaken on soil samples collected from different soil depths under different treatments before green gram seeding in the summer of 2016 (after three years of the study). In all treatments, *Euphorbia hirta*, *Amaranthus viridis*, *Celosia argentena*, *Chloris barbata*, and *Trianthema portulacum* predominated. In all three flushes, the ZT(DS)–ZT–ZT treatment had the highest weed density. Among the weed management treatments, $W_1$ with ZT(DS)–ZT–ZT had the highest weed density in the first, second, and third flushes, with 29.6%, 33.3%, and 16.7%, respectively (Figure 3c).

#### 3.1.2. Weed Density

The highest weed density was observed in the rice–wheat–green gram system when zero tillage was performed without any residue retention [ZT(DS)-ZT–ZT], followed by the T5, T3, and T1 tillage systems. The lowest weed density was observed in conventional tillage with transplanting in rice and zero tillage in wheat and green gram under the CT(T)–ZT–ZT tillage system (Table 4). Weed density studies were conducted 30 days after sowing/transplanting (DAS/T), 60 days after sowing/transplanting (DAS/T), and at harvest during all three cropping systems.

#### 3.1.3. Weed Biomass

Over the three-year experimental period, conservation tillage strategies considerably decreased weed biomass in the rice–wheat–green gram system (Table 5). In rice, $T_2$ treatment (CT(T)–ZT–ZT) had the lowest weed dry biomass (averaged across three seasons) at 30 and 60 DAS/T and at harvest, which was statistically similar to $T_1$ treatment (CT(T)–CT–fallow) and significantly better than the other conservation tillage and residue management treatments. Similarly, during the three–year trial, the lowest weed biomass was reported at 30 and 60 DAS/T with $T_1$ treatment. At harvest, CT(T)–ZT–ZT ($T_2$) had the lowest weed biomass (averaged throughout 2013–2014, 2014–2015, and 2015–2016), with values of 6.04, 6.88, and 5.98 g m$^{-2}$, respectively. Additionally, T2 treatment exhibited the lowest weed biomass at 30 and 60 DAS as well as at harvest in green gram during all three years of the study.

Across all years, the unweeded treatment ($W_3$) consistently had the largest weed biomass at 30 and 60 DAS, as well as at harvest. In contrast, the treatment $W_2$ consistently had the lowest weed biomass, which was 44.3%, 45.3%, and 33.7% lower than the biomass of $W_3$ at 30 and 60 DAS, as well as at harvest. Throughout the three-year trial, this tendency was found in both wheat and green gram crops.

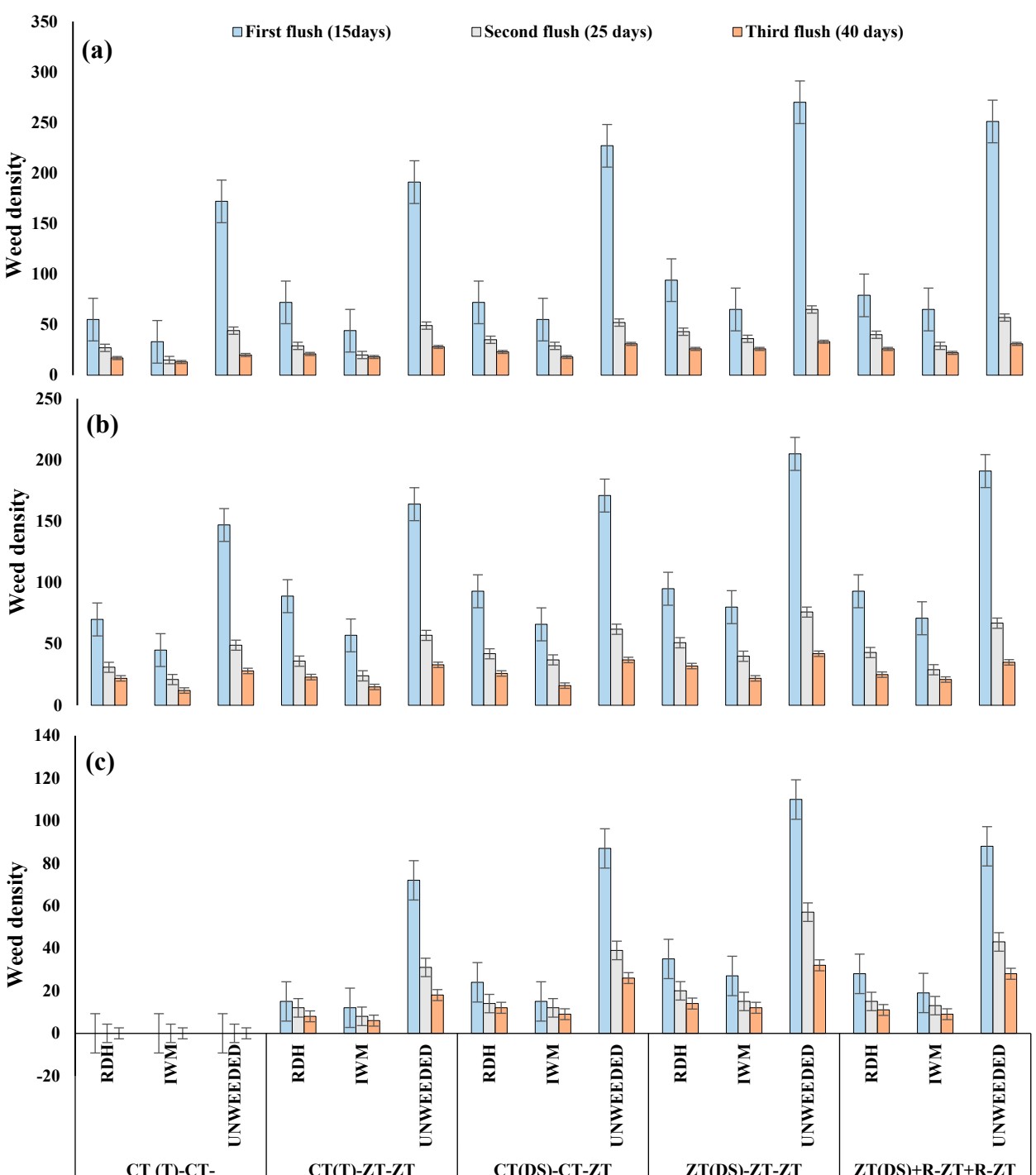

**Figure 3.** Weed seed emergence at different fluxes as affected by different tillage, residue management, and weed management methods in the wet season of 2016 (**a**), dry season of 2015 (**b**), and summer season of 2016 (**c**).

### 3.2. Crop Yield Attributes and Yield
3.2.1. Rice

Tillage and residue management had a substantial influence on yield characteristics and the yield of rice over the three-year experimental period, as shown in Table 6. The CT(DS)–CT–ZT treatment consistently had the highest number of effective tillers per square meter, with values of 44.71, 44.81, and 43.21 throughout the three years. Among the various weed control treatments, the IWM (W₂) treatment had the highest number of effective

tillers per square meter in 2014 and 2015, which was substantially higher than the other treatments. It was equivalent to $W_1$ therapy in 2016. The number of effective tillers is an essential statistic since it shows plant development and vigor. It can be influenced by various factors such as tillage practices, residue management, herbicides, and their application timing.

**Table 4.** Weed density (number m$^{-2}$) for the years 2013–2014, 2014–2015, and 2015–2016 as affected by conservation tillage and different weed management practices.

| Treatment * | Rice (Number m$^{-2}$) | | | Wheat (Number m$^{-2}$) | | | Green Gram (Number m$^{-2}$) | | |
|---|---|---|---|---|---|---|---|---|---|
| | 30 DAS/T | 60 DAS/T | At Harvest | 30 DAS | 60 DAS | At Harvest | 20 DAS | 40 DAS | At Harvest |
| | | | | 2013–2014 | | | | | |
| | | | | Tillage and residue management | | | | | |
| $T_1$ | 9.84 [c**] | 13.12 [c] | 8.16 [c] | 6.45 [e] | 24.50 [e] | 6.57 [c] | - | - | - |
| $T_2$ | 8.96 [c] | 11.86 [c] | 7.49 [c] | 7.93 [d] | 27.96 [d] | 6.04 [c] | 6.58 [d] | 35.06 [c] | 7.81 [d] |
| $T_3$ | 15.12 [b] | 20.81 [b] | 12.85 [b] | 11.26 [c] | 31.68 [c] | 11.23 [b] | 9.09 [c] | 44.34 [b] | 10.63 [c] |
| $T_4$ | 17.65 [a] | 25.65 [a] | 14.46 [a] | 13.92 [a] | 40.79 [a] | 13.58 [a] | 11.72 [a] | 49.94 [a] | 12.92 [a] |
| $T_5$ | 15.85 [b] | 22.05 [b] | 13.58 [b] | 12.21 [b] | 35.44 [b] | 11.07 [b] | 10.35 [b] | 46.76 [b] | 11.58 [b] |
| | | | | Weed management | | | | | |
| $W_1$ | 11.43 [b] | 15.21 [b] | 9.48 [b] | 7.52 b | 22.29 [b] | 8.85 [b] | 7.25 [b] | 30.53 [b] | 8.76 [b] |
| $W_2$ | 9.89 [b] | 13.85 [b] | 8.32 [b] | 6.83 b | 14.34 [c] | 7.29 [c] | 6.52 [c] | 13.10 [c] | 6.92 [c] |
| $W_3$ | 18.97 [a] | 28.11 [a] | 16.72 [a] | 16.15 a | 59.59 [a] | 15.04 [a] | 16.11 [a] | 88.45 [a] | 17.11 [a] |
| | | | | 2014–2015 | | | | | |
| | | | | Tillage and residue management | | | | | |
| $T_1$ | 10.23 [c] | 12.65 [c] | 9.41 [c] | 7.62 [e] | 23.68 [d] | 7.12 [c] | - | - | - |
| $T_2$ | 9.31 [c] | 10.45 [c] | 8.25 [c] | 9.05 [d] | 26.75 [c] | 6.88 [c] | 7.25 [d] | 33.85 [c] | 6.21 [c] |
| $T_3$ | 17.84 [b] | 21.67 [b] | 10.47 [b] | 12.47 [b] | 31.39 [b] | 10.85 [b] | 11.38 [a] | 43.21 [b] | 11.35 [b] |
| $T_4$ | 18.17 [a] | 27.04 [a] | 12.68 [a] | 14.03 [a] | 40.86 [a] | 12.50 [a] | 10.07 [b] | 48.34 [a] | 13.42 [a] |
| $T_5$ | 17.07 [b] | 22.95 [b] | 11.42 [b] | 11.19 [c] | 31.57 [b] | 10.62 [b] | 9.23 [c] | 44.74 [b] | 12.69 [a] |
| | | | | Weed management | | | | | |
| $W_1$ | 12.05 [b] | 14.41 b | 8.54 [b] | 10.05 [b] | 19.85 [b] | 9.07 [b] | 7.62 [b] | 28.62 [b] | 7.95 [b] |
| $W_2$ | 10.06 [b] | 13.99 [b] | 7.79 [b] | 8.79 [c] | 14.32 [c] | 7.95 [c] | 5.93 [c] | 12.75 [c] | 5.37 [c] |
| $W_3$ | 21.45 [a] | 28.45 [a] | 15.02 [a] | 13.77 [a] | 58.38 [a] | 11.75 [a] | 14.89 [a] | 86.24 [a] | 19.41 [a] |
| | | | | 2015–2016 | | | | | |
| | | | | Tillage and residue management | | | | | |
| $T_1$ | 9.65 [c] | 12.65 [c] | 8.25 [c] | 6.41 [e] | 23.61 [d] | 6.73 [c] | - | - | - |
| $T_2$ | 8.72 [c] | 10.45 [c] | 7.44 [c] | 8.02 [d] | 25.43 [c] | 5.98 [c] | 6.39 [d] | 33.37 [c] | 7.65 [d] |
| $T_3$ | 14.98 [b] | 21.67 [b] | 12.69 [b] | 11.24 [c] | 30.13 [b] | 11.35 [b] | 8.93 [c] | 43.00 [b] | 10.59 [c] |
| $T_4$ | 17.61 [a] | 27.04 [a] | 14.32 [a] | 14.05 [a] | 41.04 [a] | 13.48 [a] | 11.65 [a] | 47.23 [a] | 12.81 [a] |
| $T_5$ | 15.72 [b] | 22.95 [b] | 13.55 [a] | 12.18 [b] | 30.21 [b] | 10.98 [b] | 10.21 [b] | 43.26 [b] | 11.55 [b] |
| | | | | Weed management | | | | | |
| $W_1$ | 10.35 [b] | 14.41 [b] | 9.39 [b] | 7.45 [b] | 19.67 [b] | 8.73 [b] | 7.19 [b] | 27.40 [b] | 8.86 [b] |
| $W_2$ | 9.69 [b] | 13.99 [b] | 8.42 [b] | 6.91 [b] | 13.62 [c] | 7.15 [c] | 6.48 [b] | 11.44 [c] | 7.01 [c] |
| $W_3$ | 17.89 [a] | 28.45 [a] | 15.98 [a] | 15.89 [a] | 56.97 [a] | 14.95 [a] | 15.88 [a] | 86.31 [a] | 16.99 [a] |
| | | | | Mean weed density of all the three years | | | | | |
| | | | | Tillage and residue management | | | | | |
| $T_1$ | 9.90 | 12.80 | 8.60 | 6.82 | 23.93 | 6.80 | 6.74 | 34.09 | 7.22 |
| $T_2$ | 8.99 | 10.92 | 7.72 | 8.33 | 26.71 | 6.30 | 9.80 | 43.51 | 10.85 |
| $T_3$ | 15.98 | 21.38 | 12.00 | 11.65 | 31.06 | 11.14 | 11.14 | 48.50 | 13.05 |
| $T_4$ | 17.81 | 26.57 | 13.82 | 14.00 | 40.89 | 13.18 | 9.93 | 44.92 | 11.94 |
| $T_5$ | 16.21 | 22.65 | 12.85 | 11.86 | 32.40 | 10.89 | 6.74 | 34.09 | 7.22 |
| | | | | Weed management | | | | | |
| $W_1$ | 11.27 | 14.67 | 9.13 | 8.34 | 20.60 | 8.88 | 7.35 | 28.85 | 8.52 |
| $W_2$ | 9.88 | 13.94 | 8.17 | 7.51 | 14.09 | 7.46 | 6.31 | 12.43 | 6.43 |
| $W_3$ | 19.43 | 28.33 | 15.90 | 15.27 | 58.31 | 13.91 | 15.62 | 87.00 | 17.83 |

* Refer to Table 1 for treatment details. ** The means with similar letters down the column (per either tillage residue management or weed management) do not differ significantly at $p \leq 0.05$.

**Table 5.** Weed biomass (g m$^{-2}$) for the years 2013–2014, 2014–2015, and 2015–2016 as affected by conservation tillage and different weed management practices.

| Treatment * | Rice (g m$^{-2}$) | | | Wheat (g m$^{-2}$) | | | Green Gram (g m$^{-2}$) | | |
|---|---|---|---|---|---|---|---|---|---|
| | 30 DAS/T | 60 DAS/T | At Harvest | 30 DAS | 60 DAS | At Harvest | 20 DAS | 40 DAS | At Harvest |
| | | | | 2013–2014 | | | | | |
| | | | Tillage and residue management | | | | | | |
| T$_1$ | 9.84 [c**] | 13.12 [c] | 8.16 [c] | 10.82 [d] | 13.65 [d] | 12.02 [c] | - | - | - |
| T$_2$ | 8.96 [c] | 11.86 [c] | 7.49 [c] | 13.65 [c] | 15.96 [c] | 10.21 [d] | 9.81 [d] | 20.23 [c] | 10.65 [d] |
| T$_3$ | 15.12 [b] | 20.81 [b] | 12.85 [b] | 17.46 [b] | 17.18 [b] | 14.48 [b] | 15.63 [c] | 23.29 [b] | 16.12 [c] |
| T$_4$ | 17.65 [a] | 25.65 [a] | 14.46 [a] | 21.58 [a] | 21.81 [a] | 16.75 [a] | 20.25 [a] | 27.61 [a] | 20.22 [a] |
| T$_5$ | 15.85 [b] | 22.05 [b] | 13.58 [b] | 20.15 [a] | 18.41 [b] | 15.37 [b] | 18.17 [b] | 23.92 [b] | 18.91 [b] |
| | | | Weed management | | | | | | |
| W$_1$ | 17.05 [b] | 30.18 [b] | 13.42 [b] | 13.25 [b] | 11.77 [b] | 12.46 [b] | 11.67 [b] | 13.29 [b] | 11.72 [b] |
| W$_2$ | 15.82 [b] | 27.23 [b] | 12.65 [b] | 12.68 [b] | 7.73 [c] | 11.07 [c] | 9.81 [c] | 5.70 [c] | 9.65 [c] |
| W$_3$ | 28.11 [a] | 52.35 [a] | 18.62 [a] | 23.96 [a] | 32.52 [a] | 16.37 [a] | 22.18 [a] | 52.31 [a] | 21.11 [a] |
| | | | | 2014–2015 | | | | | |
| | | | Tillage and residue management | | | | | | |
| T$_1$ | 17.32 [c] | 23.45 [c] | 12.47 [b] | 9.29 [d] | 12.48 [d] | 11.63 [c] | - | - | - |
| T$_2$ | 15.48 [c] | 20.12 [c] | 9.62 [b] | 11.42 [c] | 17.22 [c] | 9.72 [d] | 8.49 [d] | 18.19 [d] | 9.15 [d] |
| T$_3$ | 23.69 [b] | 38.69 [b] | 14.78 [a] | 15.75 [b] | 18.35 [b] | 13.42 [b] | 13.65 [c] | 20.09 [c] | 14.62 [c] |
| T$_4$ | 27.15 [a] | 45.38 [a] | 16.39 [a] | 18.83 [a] | 21.64 [a] | 15.57 [a] | 17.82 [a] | 25.30 [a] | 18.75 [a] |
| T$_5$ | 28.42 [a] | 44.29 [a] | 15.22 [a] | 19.48 [a] | 19.16 [b] | 14.19 [b] | 15.07 [b] | 24.56 [b] | 17.18 [b] |
| | | | Weed management | | | | | | |
| W$_1$ | 19.47 [b] | 29.62 [b] | 19.47 [b] | 11.47 [b] | 12.32 [b] | 11.12 [b] | 10.63 [b] | 12.48 [b] | 10.74 [b] |
| W$_2$ | 16.88 [b] | 26.45 [b] | 16.88 [b] | 9.85 [c] | 8.94 [c] | 9.83 [c] | 8.45 [c] | 4.82 [c] | 8.36 [c] |
| W$_3$ | 30.88 [a] | 47.10 [a] | 30.88 [a] | 22.53 [a] | 32.05 [a] | 17.78 [a] | 22.17 [a] | 49.35 [a] | 25.66 [a] |
| | | | | 2015–2016 | | | | | |
| | | | Tillage and residue management | | | | | | |
| T$_1$ | 9.65 [c] | 12.65 [c] | 8.25 [c] | 6.41 [e] | 23.61 [d] | 6.73 [c] | - | - | - |
| T$_2$ | 8.72 [c] | 10.45 [c] | 7.44 [c] | 8.02 [d] | 25.43 [c] | 5.98 [c] | 6.39 [d] | 33.37 [c] | 7.65 [d] |
| T$_3$ | 14.98 [b] | 21.67 [b] | 12.69 [b] | 11.24 [c] | 30.13 [b] | 11.35 [b] | 8.93 [c] | 43.00 [b] | 10.59 [c] |
| T$_4$ | 17.61 [a] | 27.04 [a] | 14.32 [a] | 14.05 [a] | 41.04 [a] | 13.48 [a] | 11.65 [a] | 47.23 [a] | 12.81 [a] |
| T$_5$ | 15.72 [b] | 22.95 [b] | 13.55 [a] | 12.18 [b] | 30.21 [b] | 10.98 [b] | 10.21 [b] | 43.26 [b] | 11.55 [b] |
| | | | Weed management | | | | | | |
| W$_1$ | 16.88 [b] | 29.62 [b] | 13.62 [b] | 12.98 [b] | 12.98 [b] | 11.98 [b] | 11.72 [b] | 12.48 [b] | 11.69 [b] |
| W$_2$ | 15.69 [b] | 26.45 [b] | 12.78 [b] | 12.41 [b] | 9.33 [c] | 10.87 [b] | 9.72 [c] | 5.07 [c] | 9.56 [c] |
| W$_3$ | 27.85 [a] | 47.10 [a] | 17.98 [a] | 22.69 [a] | 31.71 [a] | 15.95 [a] | 21.36 [a] | 48.89 [a] | 20.88 [a] |
| | | | Mean weed biomass across the three years | | | | | | |
| | | | Tillage and residue management | | | | | | |
| T$_1$ | 12.27 | 16.40 | 9.62 | 8.84 | 16.58 | 10.12 | - | - | - |
| T$_2$ | 11.05 | 14.14 | 8.18 | 11.03 | 19.53 | 8.63 | 8.23 | 23.93 | 9.15 |
| T$_3$ | 17.93 | 27.05 | 13.44 | 14.81 | 21.88 | 13.08 | 12.73 | 28.79 | 13.77 |
| T$_4$ | 20.80 | 32.69 | 15.05 | 18.15 | 28.16 | 15.26 | 16.57 | 33.38 | 17.26 |
| T$_5$ | 19.9 | 29.76 | 14.11 | 17.27 | 22.59 | 13.51 | 14.48 | 30.58 | 15.88 |
| | | | Weed management | | | | | | |
| W$_1$ | 17.80 | 29.80 | 15.50 | 12.56 | 12.35 | 11.85 | 11.34 | 12.75 | 11.38 |
| W$_2$ | 16.13 | 26.71 | 14.10 | 11.64 | 8.66 | 10.59 | 9.32 | 5.19 | 9.19 |
| W$_3$ | 28.94 | 48.85 | 22.49 | 23.06 | 32.09 | 16.70 | 21.90 | 50.18 | 22.55 |

* Refer to Table 1 for treatment details. ** The means with similar letters down the column (per either tillage residue management or weed management) do not differ significantly at $p \leq 0.05$.

However, the T$_1$ treatment, which represents CT(T)–CT–fallow, consistently produced the maximum grain yield of rice (4.76, 4.81, and 4.73 tonnes per hectare) and straw yield (6.37, 6.40, and 6.40 tonnes per hectare). In terms of grain and straw yields, this treatment was statistically equivalent to the T$_2$ treatment, CT(T)–ZT–ZT, in 2014, 2015, and 2016, respectively. The T$_4$ treatment, on the other hand, resulted in the lowest rice grain production. The lowest grain and straw yields were obtained in the unweeded plots, where no weed

control techniques were employed, among the various weed management treatments. The plots that received integrated weed management (IWM) followed by $W_1$ treatment showed the highest grain yields (4.65, 4.66, and 4.58 tons per hectare). These treatments exhibited a significant increase of 34.9%, 38%, 32.1%, and 13.6% in grain and straw yields, respectively, compared to the unweeded plots (Table 6).

**Table 6.** Yield attributes and yield of rice as influenced by conservation tillage and different weed management practices.

| Treatments * | 2014 | | | 2015 | | | 2016 | | | Mean of All the Three Years | | |
|---|---|---|---|---|---|---|---|---|---|---|---|---|
| | Effective Tillers (No m$^{-2}$) | Grain Yield (t ha$^{-1}$) | Straw Yield (t ha$^{-1}$) | Effective Tillers (No m$^{-2}$) | Grain Yield (t ha$^{-1}$) | Straw Yield (t ha$^{-1}$) | Effective Tillers (No m$^{-2}$) | Grain Yield (t ha$^{-1}$) | Straw Yield (t ha$^{-1}$) | Effective Tillers (No m$^{-2}$) | Grain Yield (t ha$^{-1}$) | Straw Yield (t ha$^{-1}$) |
| | | | | | Tillage and residue management | | | | | | | |
| $T_1$ | 32.50 [b**] | 4.76 [a] | 6.37 [a] | 32.58 [d] | 4.81 [a] | 6.40 [a] | 34.98 [b] | 4.73 [a] | 6.40 [a] | 33.35 | 4.76 | 6.39 |
| $T_2$ | 37.35 [b] | 4.66 [b] | 6.07 [b] | 37.84 [c] | 4.69 [a] | 6.12 [a] | 39.54 [a] | 4.62 [a] | 6.15 [b] | 38.24 | 4.65 | 6.11 |
| $T_3$ | 44.71 [a] | 4.04 [c] | 5.66 [c] | 44.81 [a] | 4.05 [b] | 5.69 [b] | 43.21 [a] | 4.02 [b] | 5.63 [c] | 44.24 | 4.03 | 5.66 |
| $T_4$ | 37.09 [b] | 3.49 [d] | 4.81 [e] | 37.12 [c] | 3.48 [c] | 4.80 [c] | 39.63 [a] | 3.49 [c] | 4.82 [d] | 37.94 | 3.48 | 4.81 |
| $T_5$ | 41.60 [a] | 4.03 [c] | 4.92 [d] | 41.84 [b] | 4.05 [b] | 4.93 [c] | 41.28 [a] | 4.02 [b] | 4.95 [d] | 41.57 | 4.03 | 4.93 |
| | | | | | Weed management | | | | | | | |
| $W_1$ | 38.36 [b] | 4.52 [a] | 5.39 [a] | 38.80 [b] | 4.55 [b] | 5.41 [b] | 40.82 [a] | 4.52 [a] | 5.44 [b] | 39.32 | 4.53 | 5.41 |
| $W_2$ | 44.13 [a] | 4.65 [a] | 6.56 [a] | 44.20 [a] | 4.66 [a] | 6.60 [a] | 43.92 [a] | 4.58 [a] | 6.56 [a] | 44.08 | 4.63 | 6.57 |
| $W_3$ | 33.46 [c] | 3.43 [b] | 4.75 [b] | 33.50 [c] | 3.44 [c] | 4.76 [c] | 34.44 [b] | 3.43 [b] | 4.77 [c] | 33.80 | 3.43 | 4.76 |

* Refer to Table 1 for treatment details. ** The means with similar letters down the column (per either tillage residue management and Weed management) do not differ significantly at $p \leq 0.05$.

### 3.2.2. Wheat

The yield attributes of wheat were significantly influenced by conservation tillage and weed management treatments during all the years of the study (Tables 7 and 8). The number of earheads per square meter and grains per earhead were found to be the lowest in the $T_4$ treatment, with values that were 8.3% and 4.7% lower, respectively, compared to the $T_1$ treatment in the 2013–2014 season. This trend was consistently observed in the following two years of the study. Test weight, which serves as an indicator of grain quality, was highest in the $T_1$ treatment, with values of 47.68 g, 46.84 g, and 48.09 g in the three years of the study, respectively. Among the weed management treatments, the $W_2$ treatment had the highest yield qualities, including grains per earhead, the number of earheads per square meter, and test weight, whereas the unweeded plot ($W_3$) had the lowest values throughout all years (Table 7). The improved yield attributes in the weed control treatments may be attributed to the better photosynthetic efficiency of the crop, which allowed for effective weed control without causing crop damage. The weedy check plot had the lowest production characteristics due to intense crop–weed competition during the growing period. When compared to treatments with poor weed control, integrated weed management (IWM) treatments showed superior weed control and were able to obtain higher values for earheads per square meter and grains per earhead. Wheat crops with better weed control exhibited superior yield attributes due to reduced weed density. The availability of ample space, light, and nutrients for optimal crop growth and development, along with minimal interspecies competition, facilitated by the conventional tillage method in wheat, contributed to the superior yield attributes observed.

**Table 7.** Yield attributes and yield of wheat as affected by conservation tillage and different weed management practices.

| Treatment * | 2013–2014 | | | | | 2014–2015 | | | | | 2015–2016 | | | | |
|---|---|---|---|---|---|---|---|---|---|---|---|---|---|---|---|
| | Earhead (No m$^{-2}$) | Grains Earhead$^{-1}$ | Test Weight (g) | Grain Yield (t ha$^{-1}$) | Straw Yield (t ha$^{-1}$) | Earhead (No m$^{-2}$) | Grains Earhead$^{-1}$ | Test Weight (g) | Grain Yield (t ha$^{-1}$) | Straw Yield (t ha$^{-1}$) | Earhead (No m$^{-2}$) | Grains Earhead$^{-1}$ | Test Weight (g) | Grain Yield (t ha$^{-1}$) | Straw Yield (t ha$^{-1}$) |
| | | | | | | Tillage and residue management | | | | | | | | | |
| T$_1$ | 314.85 [a**] | 42.95 [b] | 47.68 [a] | 4.65 [a] | 5.42 [a] | 309.19 [a] | 41.62 [b] | 46.84 [a] | 4.73 [a] | 5.24 [a] | 314.92 [a] | 43.79 [a] | 48.09 [a] | 4.72 [a] | 5.49 [a] |
| T$_2$ | 313.41 [a] | 44.35 [a] | 46.33 [a] | 4.49 [b] | 5.21 [b] | 315.43 [a] | 42.96 [a] | 45.98 [b] | 4.46 [b] | 5.05 [b] | 314.15 [a] | 44.22 [a] | 46.70 [a] | 4.44 [b] | 5.22 [b] |
| T$_3$ | 311.77 [a] | 42.35 [b] | 45.02 [a] | 4.42 [b] | 5.19 [b] | 308.38 [a] | 40.05 [b] | 43.44 [c] | 4.39 [b] | 4.98 [b] | 312.63 [a] | 42.79 [b] | 45.44 [a] | 4.36 [c] | 5.23 [b] |
| T$_4$ | 288.90 [b] | 40.90 [c] | 41.36 [b] | 4.02 [d] | 4.76 [d] | 299.43 [b] | 39.08 [c] | 41.40 [d] | 3.99 [c] | 4.63 [c] | 290.25 [b] | 41.11 [c] | 41.61 [b] | 4.16 [e] | 4.79 [d] |
| T$_5$ | 306.28 [a] | 41.20 [c] | 43.75 [a] | 4.20 [c] | 4.99 [c] | 308.02 [a] | 38.59 [c] | 43.88 [c] | 4.14 [c] | 4.76 [c] | 307.02 [a] | 41.71 [c] | 43.97 [a] | 4.24 [d] | 5.01 [c] |
| | | | | | | Weed management | | | | | | | | | |
| W$_1$ | 311.46 [a] | 43.51 [a] | 45.14 [a] | 4.68 [b] | 5.58 [b] | 317.11 [a] | 40.77 [b] | 44.17 [b] | 4.75 [b] | 5.47 [a] | 312.93 [a] | 43.63 [a] | 45.32 [a] | 4.74 [a] | 5.64 [a] |
| W$_2$ | 311.20 [a] | 44.50 [a] | 47.96 [a] | 4.89 [a] | 5.78 [a] | 315.86 [a] | 42.58 [a] | 47.74 [a] | 4.91 [a] | 5.48 [a] | 310.99 [a] | 44.97 [a] | 48.50 [a] | 4.91 [a] | 5.83 [a] |
| W$_3$ | 298.46 [a] | 39.05 [b] | 41.38 [b] | 3.49 [c] | 3.99 [c] | 291.29 [b] | 38.04 [c] | 41.02 [c] | 3.37 [c] | 3.84 [b] | 299.46 [b] | 39.58 [b] | 41.66 [b] | 3.50 [b] | 3.98 [b] |

* Refer to Table 1 for treatment details. ** The means with similar letters down the column (per either tillage residue management and weed management) do not differ significantly at $p \leq 0.05$.

**Table 8.** Mean of yield attributes and yield of wheat for the three years as influenced by conservation tillage and different weed management practices.

| Treatment * | Earhead (No m$^{-2}$) | Grains Earhead$^{-1}$ | Test Weight (g) | Grain Yield (t ha$^{-1}$) | Straw Yield (t ha$^{-1}$) |
|---|---|---|---|---|---|
| | | Tillage and residue management | | | |
| T$_1$ | 312.98 | 42.78 | 47.53 | 4.70 | 5.38 |
| T$_2$ | 314.33 | 43.84 | 46.33 | 4.46 | 5.16 |
| T$_3$ | 310.92 | 41.73 | 44.63 | 4.39 | 5.13 |
| T$_4$ | 292.86 | 40.36 | 41.45 | 4.05 | 4.72 |
| T$_5$ | 307.10 | 40.50 | 43.86 | 4.19 | 4.92 |
| | | Weed management | | | |
| W$_1$ | 313.83 | 42.63 | 44.87 | 4.72 | 5.56 |
| W$_2$ | 312.68 | 44.01 | 48.06 | 4.90 | 5.69 |
| W$_3$ | 296.40 | 38.89 | 41.35 | 3.45 | 3.93 |

* Refer to Table 1 for treatment details.

The $T_1$ treatment consistently produced the highest grain production (4.65 t/ha, 4.73 t/ha, and 4.72 t/ha averaged over three years), which was considerably superior to the other treatments. The $T_4$ treatment, on the other hand, had the lowest grain yield. This tendency was mirrored in straw production, with $T_1$ yielding 13.9% and 9.3% more straw (averaged over three years) than $T_4$ and $T_5$, respectively (Table 7). The lower yields in the zero-tillage treatments can be attributed to a lack of available growth resources as a result of increased weed competition, which hampered wheat crop growth and development, resulting in poorer yield-attributing traits. In the year 2013–2014, the greatest grain yield (4.89 t/ha) and straw yield (5.78 t/ha) was recorded in $W_2$ (4.68 t/ha, 5.58 t/ha), which was considerably superior to the other weed management treatments (Table 7). The greatest grain production of 4.91 t/ha was observed in the second year of the trial under the IWM treatment, which was considerably superior to all other weed management regimens. $W_2$ had the highest straw yield (5.48 t/ha), which was comparable to $W_1$ (5.47 t/ha) and much higher than $W_3$ (3.84 t/ha). Similarly, $W_2$ had the highest grain and straw yield, which was comparable to $W_1$ but much higher than $W_3$.

### 3.2.3. Green Gram

Yield attributes and yield of summer green gram were significantly influenced by conservation tillage and various weed management practices (Tables 9 and 10). The findings revealed that there was no significant difference in the number of pods per plant in 2014, while the $T_2$ treatment had the maximum number of pods per plant in 2014 and 2015, with values of 22.86 and 22.61 pods per plant, respectively. Furthermore, the CT(DS)–CT–ZT ($T_3$) treatment consistently produced the most seeds per pod for three years. In the first and second years of the study, there was no significant difference in test weight among the different tillage and residue management treatments. However, in the third year (2016), the $T_5$ treatment recorded the highest test weight, which was comparable to all treatments except $T_4$. Among the weed management treatments, the highest values for yield-attributing characteristics were observed in the $W_2$ treatment, while the lowest values were found in the $W_3$ treatment.

The $T_3$ treatment, which included CT(DS)–CT–ZT, produced the greatest grain yield of green gram (1.48 t/ha), and it was considerably superior to the other treatments. Similarly, $T_3$ produced the maximum straw yield of 2.60 t/ha, whereas $T_4$ produced the lowest straw yield (averaged over three years) of 1.85 t/ha. Throughout the years, the IWM ($W_2$) herbicidal treatment consistently produced the highest grain and straw yields. $W_3$ treatment, on the other hand, had a 32.0% and 32.7% lower grain and straw yield (averaged over three years) than $W_1$ treatment (Table 9). These differences in yield can be attributed to variations in the production of pods per plant and seeds per pod, which ultimately contributed to an increased overall yield. The significant reduction in weed competition, achieved through effective weed management, played a crucial role in promoting the overall growth of the crop at all stages of observation, thus positively impacting the final yield. The average yield of rice, wheat, and green gram for all the years is presented in Figure 4.

**Table 9.** Yield attributes and yield of green gram as affected by conservation tillage and different weed management practices.

| Treatment * | 2013–2014 | | | | | 2014–2015 | | | | | 2015–2016 | | | | |
|---|---|---|---|---|---|---|---|---|---|---|---|---|---|---|---|
| | Number of Pods Plants$^{-1}$ | Number of Seeds Pod$^{-1}$ | Test Weight (g) | Grain Yield (t ha$^{-1}$) | Straw Yield (t ha$^{-1}$) | Number of Pods Plants$^{-1}$ | Number of Seeds Pod$^{-1}$ | Test Weight (g) | Grain Yield (t ha$^{-1}$) | Straw Yield (t ha$^{-1}$) | Number of Pods Plants$^{-1}$ | Number of Seeds Pod$^{-1}$ | Test Weight (g) | Grain Yield (t ha$^{-1}$) | Straw Yield (t ha$^{-1}$) |
| | *Tillage and residue management* | | | | | | | | | | | | | | |
| T$_1$ | - | - | - | - | - | - | - | - | - | - | - | - | - | - | - |
| T$_2$ | 22.36 [a**] | 8.22 [b] | 54.25 [a] | 1.52 [b] | 2.34 [a] | 22.86 [a] | 8.65 [b] | 53.36 [a] | 1.14 [b] | 2.35 [b] | 22.61 [a] | 8.36 [b] | 54.15 [b] | 1.17 [c] | 2.40 [b] |
| T$_3$ | 22.22 [a] | 9.53 [a] | 54.01 [a] | 1.82 [a] | 2.57 [a] | 21.92 [b] | 10.04 [a] | 53.16 [a] | 1.31 [a] | 2.60 [a] | 22.44 [a] | 8.90 [a] | 54.39 [b] | 1.31 [a] | 2.64 [a] |
| T$_4$ | 20.54 [b] | 7.61 [b] | 51.43 [b] | 1.35 [c] | 1.84 [b] | 20.56 [c] | 7.75 [c] | 50.10 [b] | 0.94 [c] | 1.88 [c] | 20.57 [b] | 7.68 [c] | 51.56 [c] | 0.97 [d] | 1.85 [c] |
| T$_5$ | 21.59 [a] | 8.51 [b] | 54.75 [a] | 1.64 [b] | 2.56 [a] | 21.48 [b] | 8.70 [b] | 52.53 [a] | 1.30 [a] | 2.53 [a] | 21.68 [a] | 8.63 [a] | 54.76 [a] | 1.24 [b] | 2.57 [b] |
| | *Weed management* | | | | | | | | | | | | | | |
| W$_1$ | 21.71 [b] | 8.46 [b] | 53.51 [b] | 1.55 [b] | 2.45 [b] | 21.82 [b] | 8.92 [b] | 51.59 [b] | 1.24 [a] | 2.42 [b] | 21.84 [b] | 8.68 [a] | 53.72 [b] | 1.25 [a] | 2.53 [b] |
| W$_2$ | 24.12 [a] | 9.30 [a] | 56.65 [a] | 1.83 [a] | 2.71 [a] | 24.01 [a] | 9.64 [a] | 55.67 [a] | 1.37 [a] | 2.74 [a] | 24.29 [a] | 8.78 [a] | 56.92 [a] | 1.39 [a] | 2.73 [a] |
| W$_3$ | 19.20 [c] | 7.64 [c] | 50.67 [c] | 1.27 [c] | 1.83 [c] | 19.14 [c] | 7.79 [c] | 49.60 [c] | 0.91 [b] | 1.85 [c] | 19.35 [c] | 7.73 [b] | 50.51 [c] | 0.93 [b] | 1.82 [c] |

* Refer to Table 1 for treatment details. ** The means with similar letters down the column (per either tillage residue management and weed management) do not differ significantly at $p \leq 0.05$.

**Table 10.** Mean of yield attributes and yield of green gram across the three years as affected by conservation tillage and different weed management practices.

| Treatment * | Number of Pods Plants$^{-1}$ | Number of Seeds Pod$^{-1}$ | Test Weight (g) | Grain Yield (t ha$^{-1}$) | Straw Yield (t ha$^{-1}$) |
|---|---|---|---|---|---|
| | *Tillage and residue management* | | | | |
| T$_1$ | - | - | - | - | - |
| T$_2$ | 22.61 | 8.41 | 53.92 | 1.27 | 2.36 |
| T$_3$ | 22.19 | 9.49 | 53.85 | 1.48 | 2.60 |
| T$_4$ | 20.55 | 7.68 | 51.03 | 1.08 | 1.85 |
| T$_5$ | 21.58 | 8.61 | 54.01 | 1.39 | 2.55 |
| | *Weed management* | | | | |
| W$_1$ | 21.79 | 8.68 | 52.94 | 1.34 | 2.46 |
| W$_2$ | 24.14 | 9.24 | 56.41 | 1.53 | 2.72 |
| W$_3$ | 19.23 | 7.72 | 50.26 | 1.03 | 1.83 |

* Refer to Table 1 for treatment details.

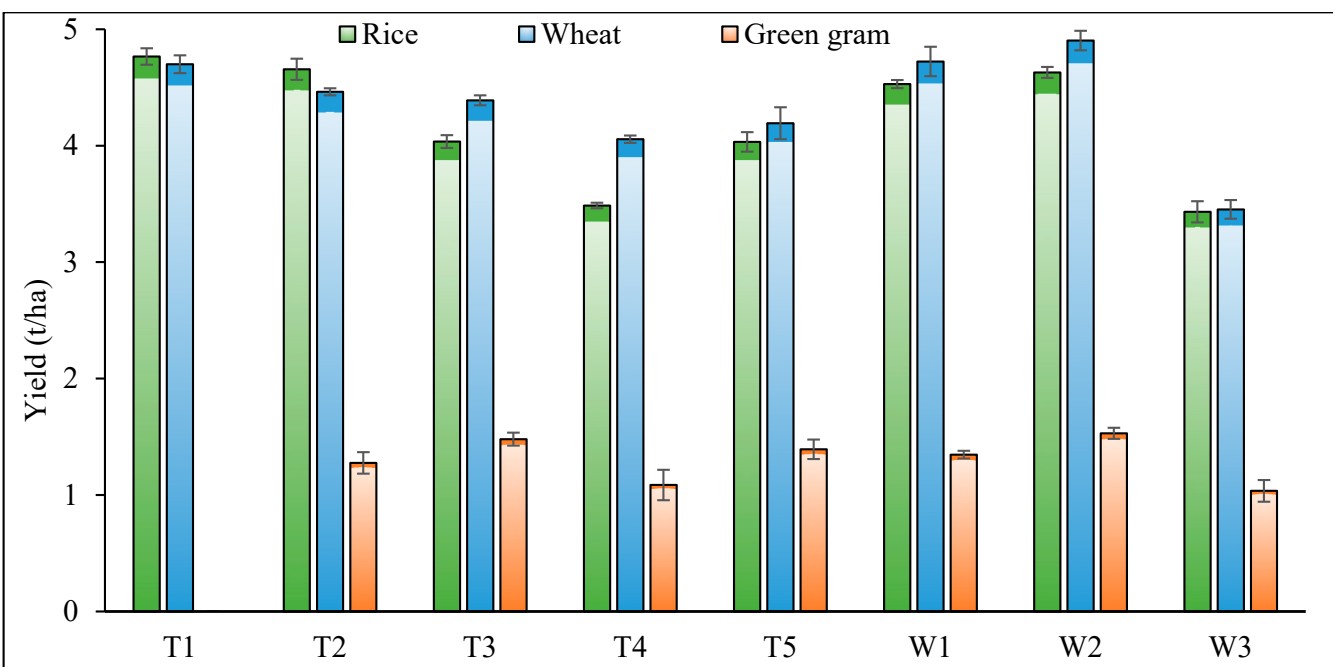

**Figure 4.** Mean yield (3 years) of rice, wheat, and green gram.

### 3.3. Soil Health Parameters

#### 3.3.1. Soil Chemical Properties

There were significant differences in soil pH due to the adoption of various tillage and residue management practices. The maximum reduction in pH (7.32) from the initial value (8.04) was observed in the $T_5$ treatment (ZT(DS) + R–ZT + R–ZT). Conversely, the highest increase in pH from the initial value was found in $T_1$ treatment (CT(T)–CT–fallow) (Table 11). Among the weed management treatments, the unweeded treatment ($W_3$) resulted in the maximum reduction in pH (7.91).

**Table 11.** Chemical properties of the post-harvest soil under the rice–wheat–green gram cropping system as influenced by conservation tillage and different weed management practices.

| Treatment * | pH | Organic Carbon (%) | Available N (kg ha$^{-1}$) | Available P$_2$O$_5$ (kg ha$^{-1}$) | Available K$_2$O (kg ha$^{-1}$) |
|---|---|---|---|---|---|
| | | Tillage and residue management | | | |
| $T_1$ | 8.85 [a]** | 0.46 [c] | 250.52 [a] | 43.87 [a] | 281.34 [a] |
| $T_2$ | 7.50 [b] | 0.53 [b] | 233.09 [b] | 44.95 [a] | 284.63 [a] |
| $T_3$ | 8.19 [a] | 0.58 [a] | 251.15 [a] | 50.56 [a] | 279.58 [a] |
| $T_4$ | 8.17 [a] | 0.54 [b] | 250.89 [a] | 47.88 [a] | 283.25 [a] |
| $T_5$ | 7.32 [b] | 0.59 [a] | 251.18 [a] | 50.05 [a] | 266.69 [b] |
| | | Weed management | | | |
| $W_1$ | 7.93 [a] | 0.47 [b] | 240.05 [a] | 49.47 [a] | 284.35 [a] |
| $W_2$ | 8.34 [a] | 0.52 [a] | 241.22 [a] | 50.38 [a] | 286.59 [a] |
| $W_3$ | 7.91 [a] | 0.55 [a] | 239.35 [a] | 44.05 [b] | 268.87 [a] |

\* Refer to Table 1 for treatment details. ** The means with similar letters down the column (per either tillage residue management or weed management) do not differ significantly at $p \leq 0.05$.

The organic carbon (%) content of the soil after harvest was also measured and presented in Table 11. The treatments had a considerable impact on the soil's organic carbon concentration. At the start of the trial in 2013 (before wheat seeding), the original soil organic carbon concentration was 0.51%. The $T_5$ treatment showed the greatest increase in soil organic carbon content (0.59%) above the original value. The $T_1$ treatment (CT(T)–CT–fallow) had the greatest loss in soil organic carbon (0.41%) from the starting value.

In terms of weed management treatments, the $W_3$ treatment had the highest increase in organic carbon (0.55%), which was equivalent to the $W_2$ treatment (0.52%) and 17% greater than the $W_1$ treatment.

The analysis of available N, P, and K (kg ha$^{-1}$) was conducted after the harvest of the rice–wheat–green gram cropping system in the year 2016, specifically after the green gram harvest, and the results are presented in Table 11. Among the different tillage and residue management treatments, $T_5$ had the highest available N content of 251.18 kg ha$^{-1}$, representing a 2.5% increase over the original value. $P_2O_5$ and $K_2O$ levels were greatest in $T_3$ and $T_2$, with values of 50.56 kg ha$^{-1}$ and 284.63 kg ha$^{-1}$, respectively. Conversely, the minimum available N content was recorded in $T_2$ (233.09 kg ha$^{-1}$), while the lowest available $P_2O_5$ and $K_2O$ levels were found in T1 and T5, with values of 40.65 kg ha$^{-1}$ and 266.69 kg ha$^{-1}$, respectively (Table 11). Notably, T5 exhibited a 7.7% higher available N content compared to $T_2$. Additionally, $T_3$ and $T_2$ demonstrated a 15.2% and 6.7% increase in available $P_2O_5$ and $K_2O$, respectively, over $T_1$ and $T_5$ after the three-year study period. Similarly, the weed management treatment $W_2$ resulted in the maximum N, $P_2O_5$, and $K_2O$ content in the post-harvest soil, measuring 241.22 kg ha$^{-1}$, 50.38 kg ha$^{-1}$, and 286.59 kg ha$^{-1}$, respectively, and it was significantly superior to the other weed management treatments.

### 3.3.2. Soil Biological Properties

Among the tillage and residue management treatments, soil biological properties differed significantly. The highest number of Azotobacter (104 cfu g$^{-1}$ soil), with a value of 3.95, was found in the CT(DS)–CT–ZT treatment, which was 32.5% higher compared to the CT(T)–CT–fallow treatment. Similarly, $T_5$ exhibited the maximum values for Total Pseudomonas (105 cfu g$^{-1}$ soil), Total phosphate solubilizing bacteria (PSB) (105 cfu g$^{-1}$ soil), percentage of P solubilized by Pseudomonas and Bacillus (105 cfu g$^{-1}$ soil), and $CO_2$ evolution (mg kg$^{-1}$), with increases of 40.4%, 34.3%, 32.7%, 49.1%, 54%, and 56.4%, respectively, compared to $T_1$ (Table 12). The biological properties of the soil vary significantly among the weed management treatments. The highest numbers of Azotobacter, total Pseudomonas, total PSB, and Bacillus (3.81, 6.54, 9.42, and 4.95, respectively) were found in $W_3$. Additionally, the percentage of P solubilized by Pseudomonas and by Bacillus was 14.5% and 19.9% higher, respectively, in W3 compared to W2 (Table 12). The lowest $CO_2$ evolution (mg kg$^{-1}$) was observed in $W_1$ (74.87), followed by $W_2$ (75.29) and $W_3$ (82.88).

**Table 12.** Biological properties of post-harvest soil under the rice–wheat–green gram cropping system as affected by conservation tillage and different weed management practices.

| Treatment * | Azotobacter ($10^4$ cfu g$^{-1}$ Soil) | Total Pseudomonas ($10^5$ cfu g$^{-1}$ Soil) | Total PSB ($10^5$ cfu g$^{-1}$ Soil) | % of P Solubilized by Pseudomonas | Bacillus ($10^5$ cfu g$^{-1}$ Soil) | % of P Solubilized by Bacillus | $CO_2$ Evolution (mg kg$^{-1}$) |
|---|---|---|---|---|---|---|---|
| | | | Tillage and residue management | | | | |
| $T_1$ | 2.98 [b**] | 4.63 [b] | 7.11 [a] | 19.53 [b] | 4.05 [a] | 15.68 [b] | 63.78 [d] |
| $T_2$ | 3.31 [a] | 4.95 [b] | 8.22 [a] | 23.75 [a] | 4.13 [a] | 21.32 [a] | 69.95 [c] |
| $T_3$ | 3.95 [a] | 5.57 [b] | 8.89 [a] | 23.12 [a] | 4.59 [a] | 18.26 [b] | 69.52 [c] |
| $T_4$ | 3.45 [a] | 5.03 [b] | 9.39 [a] | 23.51 [a] | 4.35 [a] | 20.89 [a] | 83.56 [b] |
| $T_5$ | 3.93 [a] | 8.68 [a] | 9.55 [a] | 25.92 [a] | 6.24 [a] | 23.38 [a] | 99.78 [a] |
| | | | Weed management | | | | |
| $W_1$ | 3.31 [a] | 5.58 [b] | 7.88 [b] | 22.40 [a] | 4.46 [a] | 19.19 [a] | 74.87 [b] |
| $W_2$ | 3.43 [a] | 5.26 [b] | 8.57 [a] | 22.04 [a] | 4.18 [a] | 18.59 [b] | 75.29 [b] |
| $W_3$ | 3.81 [a] | 6.54 [a] | 9.42 [a] | 25.25 [a] | 4.95 [a] | 22.29 [a] | 82.88 [a] |

* Refer to Table 1 for treatment details. ** The means with similar letters down the column (per either tillage residue management or weed management) do not differ significantly at $p \leq 0.05$.

### 3.4. Economics

The selection of tillage, residue, and weed management practices is influenced by the economic returns they offer, as farmers prioritize higher returns per unit area, time, and investment. The cost of cultivation varied significantly depending on the tillage, residue,

and weed management methods. In the rice–wheat–green gram cropping system, the total cost of production followed this order: $T_5 > T_2 > T_4 > T_3 > T_1$; $W_2 > W_1 > W_3$ (Table 13). At the system level, the highest cost of production was incurred in the ZT(DS) + R-ZT + R-ZT treatment (INR 68397 or USD 1121.4, 1066.5, 1019 ha$^{-1}$ during the three years, respectively) among the tillage and residue management treatments, and in the IWM treatment (INR 77664 or USD 1273.4, 1211, 1157 ha$^{-1}$ during the three years, respectively) among the weed management treatments.

The highest gross returns (INR 188841 or USD 3096.2, INR 184231 or USD 2872.7, INR 220322 or USD 3279.5 ha$^{-1}$ during the three years, respectively) were recorded under the T2 treatment, CT(T)–ZT–ZT, in tillage and residue management, which were 19%, 21%, and 17.7% higher compared to T4 during the 2013–2014, 2014–2015, and 2015–2016 rice–wheat–green gram system, respectively (Table 13). Among the weed management practices, the highest gross return was recorded under the $W_2$ treatment during all the years.

The highest net return (INR 120628 or USD 1977.8, INR 116017 or USD 1809, INR 152109 or USD 2266.2 ha$^{-1}$ during the three years, respectively) was also recorded under the $T_2$ treatment, CT(T)–ZT–ZT, in tillage and residue management, which were 33.6%, 37.6%, and 27.7% higher compared to $T_4$ during the 2013–2014, 2014–2015, and 2015–2016 rice–wheat–green gram system, respectively (Table 13). Among the weed management practices, the highest net return was recorded under the $W_2$ treatment during all the years.

In the rice–wheat–green gram production system, the order of the benefit-to-cost (B:C) ratio among the tillage and residue management treatments was $T_2$ (2.90) = $T_3$ (2.88) > $T_1$ (2.79) > $T_5$ (2.77) > $T_4$ (2.44) (averaged over three years). Similarly, the highest B:C ratio was found under the $W_1$ treatment, which involved recommended herbicides. The higher economic yield under $W_1$ was attributed to the reduction in competition from weeds during the most critical stages of the crop-weed competition.

**Table 13.** Economics of the rice–wheat–green gram cropping system as affected by conservation tillage and different weed management practices.

| Treatment * | 2013–2014 | | | | 2014–2015 | | | | 2015–2016 | | | |
|---|---|---|---|---|---|---|---|---|---|---|---|---|
| | Cost of Cultivation (INR ha$^{-1}$) | Gross Returns (INR ha$^{-1}$) | Net Returns (INR ha$^{-1}$) | B:C Ratio | Cost of Cultivation (INR ha$^{-1}$) | Gross Returns (INR ha$^{-1}$) | Net Returns (INR ha$^{-1}$) | B:C Ratio | Cost of Cultivation (INR ha$^{-1}$) | Gross Returns (INR ha$^{-1}$) | Net Returns (INR ha$^{-1}$) | B:C Ratio |
| Tillage and residue management | | | | | | | | | | | | |
| $T_1$ | 55820 | 145504 [d**] | 94384 [c] | 2.75 [a] | 55820 | 146852 [c] | 95732 [c] | 2.63 [b] | 55820 | 166956 [d] | 115836 [d] | 2.99 [c] |
| $T_2$ | 68213 | 188841 [a] | 120628 [a] | 2.77 [a] | 68213 | 184231 [a] | 116017 [a] | 2.70 [a] | 68213 | 220322 [a] | 152109 [a] | 3.24 [a] |
| $T_3$ | 67597 | 184905 [a] | 117308 [a] | 2.76 [a] | 67597 | 180782 [a] | 113186 [a] | 2.67 [a] | 67597 | 216816 [b] | 149219 [a] | 3.23 [a] |
| $T_4$ | 67897 | 158616 [c] | 90270 [c] | 2.34 [c] | 67897 | 152200 [c] | 84304 [d] | 2.24 [d] | 67897 | 187035 [d] | 119139 [c] | 2.76 [d] |
| $T_5$ | 68397 | 179312 [b] | 110916 [b] | 2.64 [b] | 68397 | 175290 [b] | 106894 [b] | 2.56 [c] | 68397 | 211774 [c] | 143378 [b] | 3.11 [b] |
| Weed management | | | | | | | | | | | | |
| $W_1$ | 60620 | 183591 [b] | 122971 [a] | 3.04 [a] | 60620 | 180771 [b] | 120151 [a] | 2.98 [a] | 60620 | 214635 [b] | 154015 [a] | 3.55 [a] |
| $W_2$ | 77664 | 193351 [a] | 115687 [b] | 2.51 [b] | 77664 | 189933 [a] | 112269 [b] | 2.45 [b] | 77664 | 226149 [a] | 148485 [a] | 2.92 [b] |
| $W_3$ | 55650 | 137366 [c] | 81716 [c] | 2.47 [b] | 55650 | 132909 [c] | 77259 [c] | 2.39 [b] | 55650 | 160959 [c] | 105309 [b] | 2.90 [b] |

* Refer to Table 1 for treatment details. ** The means with similar letters down the column (per either tillage residue management or weed management) do not differ significantly at $p \le 0.05$.

## 4. Discussion

### 4.1. Weed Dynamics

Numerous variables influence how much tillage affects the amount of the weed seed bank [50]. Since tillage has a diminutive effect [51] on reducing [52] or increasing [53] weed seed bank density, empirical investigations produce contradictory consequences. The results of plentiful research indicate that the weed species governs how the weed seed bank reacts to tillage [32]. These studies also specified that the complex interplay between weather, the span of the experiment, and long-term field history affects how the weed seed bank responds to tillage. Although it can be time-consuming and challenging to assess, the initial state and distribution of the weed seed bank have a significant impact on study outcomes. Tillage reallocates seeds all over the soil profile, regardless of the texture and structure of the soil [54]. A higher percentage of ZT seedbanks will germinate than CT

regime seedbanks [55], which have generalized patterns of seed distribution, because ZT seeds infiltrate the soil via very slow processes through thin cracks, diversified macro-fauna and freeze-dry cycles [56], resulting in an accumulation of weed seeds (60–90%) in the top 5 cm of the soil [57]. A higher density of weeds in zero-till rice was also reported by Nichols et al. [58]. ZT involves minimal or no soil disturbance, leaving the weed seeds undisturbed and closer to the soil surface, allowing the weed seeds to remain viable and readily available for germination, leading to higher weed densities [58]. Also, weed seeds are not buried as deeply into the soil in ZT as they would be with CT, and this shallow seed placement provides favorable conditions for weed seed germination and emergence [50,59], contributing to increased weed density. It might also be possible that CT can cause physical damage to weed seeds, resulting in decreased viability and germination potential, whereas ZT practices typically do not subject weed seeds to the same level of physical disturbance, allowing a higher percentage of seeds to remain viable, leading to increased weed densities.

Residue-laden treatments have shown better weed control over clean cultivation owing to the prevention of weeds from germinating and may facilitate higher seed predation due to favorable conditions for soil macro-fauna, such as ants and beetles [60,61]. Residue retention is beneficial for reducing weed populations in crop fields because it acts as a physical barrier [62], preventing weed seeds from reaching the soil surface and germinating. At the same time, the residue layer hinders weed seedling emergence by limiting light penetration and creating an unfavorable environment for weed growth. Additionally, the decaying crop residues release allelochemicals that possess herbicidal properties, inhibiting weed germination and growth [63,64]. By retaining crop residues, weed populations can be effectively suppressed, leading to improved weed control and higher yields in rice cultivation.

The continuous use of single herbicides or single weed management practices will lead to undesirable phenomena like herbicide resistance, weed shift, and many more issues [65]. Combining cultural practices such as crop rotation, like in the case of the present study involving summer green gram [66], and residue management with mechanical methods such as intercropping and manual weeding effectively suppresses weeds, optimizing resource utilization by crops [67]. This has ultimately resulted in a reduction in weed density as well as weed biomass in IWM treatment.

*4.2. Crop Yield*

Based on the findings of the present study, it can be inferred that the yield in ZT rice plots was comparatively lower than in CT, which aligns with the earlier research conducted by Alam et al. [68]. The decrease in yield observed in direct-seeded rice (DSR) can be attributed to various factors. These include soil sickness caused by nutrient unavailability compared to conventional tillage (CT), vigorous weed growth favored by alternating wet and dry cycles and the absence of standing water, moisture stress due to higher percolation rates, potential stress from nematodes and rice mealybugs, and increased spikelet sterility, all of which pose significant challenges to achieving high grain yields in rice cultivation. However, effective management of both biotic and abiotic stresses, such as controlling weed growth, nematode infestation, and leaf miner attacks, can greatly alleviate these yield losses in DSR. On the other hand, in CT rice cultivation, the practice of puddling provides several advantages. It enhances weed control by creating an unfavorable environment for their growth, reduces water and nutrient loss through deep percolation, facilitates the rapid establishment of rice seedlings, and improves nutrient availability by utilizing the redox potential phenomenon in waterlogged soil.

Being an integral part of conservation agricultural practices, ZT has gained popularity in wheat cultivation due to its potential benefits such as soil moisture conservation, reduced erosion, and cost savings [37,69]. However, despite these advantages, zero-tilled wheat systems sometimes experience a decline in yield compared to conventional tillage methods, attributed to numerous factors, including soil compaction, increased weed competition [32], challenges associated with residue management [62], nutrient imbalances [70], and disease

and pest pressure. ZT often leads to increased soil compaction due to the absence of tillage operations that help alleviate compaction, hindering root growth, reducing nutrient uptake, and limiting water infiltration, resulting in decreased plant growth and, ultimately, lowering yields. These factors aggravated each other further because ZT experienced greater weed pressure as compared to CT methods. The absence of tillage disrupts weed seed burial and exposes them to favorable germination conditions, leading to amplified weed competition for resources such as nutrients, light, and water that can significantly impact the wheat yield [55]. On the contrary, CT provides better weed control due to tillage activity at the time of crop establishment, leading to better yield-attributing characteristics and yield [59]. Moreover, in ZT, crop residues are left on the soil surface, which might create challenges for the germination of emerging wheat crops by impeding seed-to-soil contact [71], hindering seedling emergence, and reducing early plant vigor, leading to a lesser number of tillers and other yield-attributing characteristics and, in turn, affecting the yield. ZT may also experience imbalances in nutrient availability and uptake [72], inferred due to the accumulation of crop residues on the soil surface leading to nutrient immobilization [73,74], making essential nutrients like N and P less accessible to the growing wheat plants [75]. Additionally, without tillage, nutrient stratification may occur [76], with nutrients concentrated in the surface layers and limiting their availability in the lower root zone. Also, in many instances, the residue was the chief source of promulgation of certain diseases and pests by acting as an alternate host and habitat for varied plant pathogens and pests [77], potentially leading to higher disease pressure and insect infestations and resulting in reduced stand establishment, poor plant health, and yield losses [78].

The higher yield in summer green gram following CT rice and wheat can be attributed to a combination of interconnected factors encompassing residue decomposition and nutrient availability, effective weed suppression, enhanced soil aeration and root penetration, better pest and disease management, and soil moisture conservation [79,80]. CT incorporates crop residues, promoting their decomposition and releasing nutrients that are readily available for the subsequent crop, enhancing the growth, yield, and yield-attributing characteristics of summer green gram [81]. Additionally, CT aids in effective weed control by burying weed seeds and disrupting their germination, reducing competition and providing the summer green gram crop with a competitive advantage [50,59]. Improved soil aeration and root penetration achieved through CT practices further support nutrient uptake and overall crop performance. Furthermore, CT disrupts pest life cycles, reduces disease incidence, and buries pests, pathogens, and infected crop residues, thereby minimizing pest and disease pressure and safeguarding the yield potential of further crops in sequence [77]. As CT operations break up surface soil crusts, facilitating better water infiltration and ensuring optimal moisture availability for summer green gram germination, establishment, and growth [82]. In contrast, ZT systems exhibit slower residue decomposition rates, potentially leading to delayed nutrient release, resulting in limited nutrient availability and reduced crop productivity [83]. Moreover, the presence of undisturbed crop residues in ZT systems can foster weed growth, increase weed competition, and hinder the yield potential of subsequent crops. Also, the compact soil layers in ZT can impede root growth, nutrient accessibility, and overall crop performance.

Conservation tillage in conjugation with integrated weed management (IWM) practices, i.e., $W_2$ in the rice–wheat–green gram sequence, provides scientifically substantiated benefits, including higher yields. This might be due to low weed density during the initial crop growth stages. The timely application of herbicides and further control of later germinated weeds by the supplemented intercultural operation followed by hand weeding caused a reduction in weed competition, leading to increased photosynthetic activity, and biomass accumulation [84,85]. The correlation among different parameters of rice and wheat across the years has been presented in Figures 5 and 6.

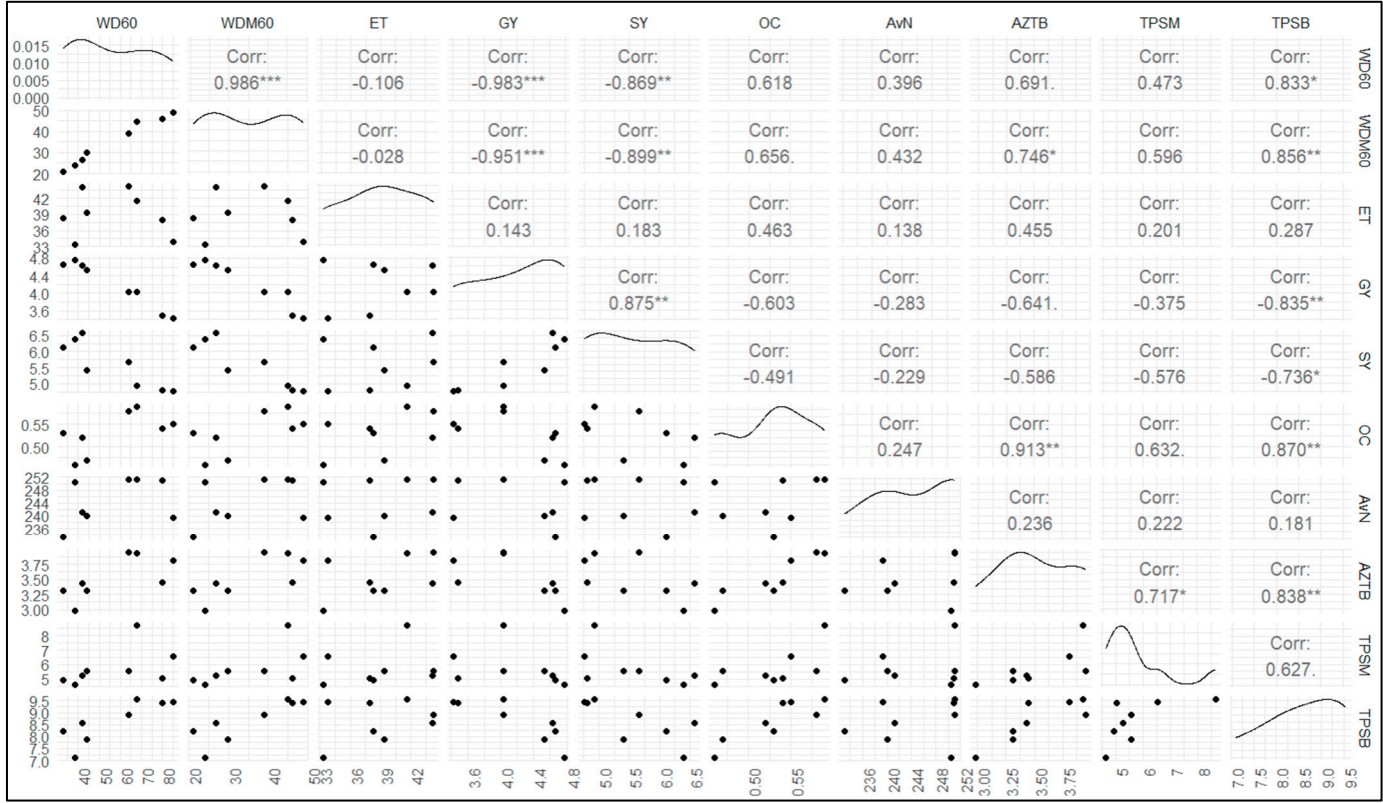

**Figure 5.** Correlation panel graph among different parameters of rice cultivation across the years (mean of 3 years, WD60: weed density at 60 days after sowing (60DAS), WDM60: weed dry matter at 60DAS, ET: number of effective tillers, GY: grain yield, SY: Straw yield, OC: organic carbon in soil, AvN: available nitrogen in soil, AZTB: Azotobacter population, TPSM: total Pseudomonas population, TPSB: total phosphate solubilizing bacteria population. * Significance at $p < 0.05$, ** significance at $p < 0.01$, *** significance at $p < 0.001$).

*4.3. Soil Properties*

The maximum upsurge in SOC content (0.59%) from the initial value was observed in $T_5$, where the ZT practices offer several interlinked reasons for higher OC content and increased nutrient availability in comparison to CT soil. Firstly, ZT minimizes soil disturbance, preserving soil organic matter (SOM) and preventing its oxidation, resulting in higher OC levels [86,87]. The presence of crop residues on the soil surface in ZT systems further contributes to OC accumulation by providing continuous organic material that gradually decomposes, which enhances the SOC content and positively influences soil health and nutrient availability [88,89]. Moreover, ZT promotes nutrient retention by reducing leaching through the physical barrier created by crop residues, improving nutrient availability, mainly N, P, and K, for plant uptake [90,91]. Additionally, the undisturbed soil structure and increased OC content in ZT soil foster a diverse and active microbial community that plays a vital role in nutrient cycling and mineralization [92], converting SOM into plant-available forms ([93]. The improved soil structure and stability of soil aggregates in ZT systems protect SOM and nutrients from degradation [94] and promote root exploration and nutrient uptake, enhancing nutrient availability [95]. Furthermore, ZT practices reduce erosion by maintaining soil surface cover, which prevents the loss of SOM and nutrients through wind or water erosion [96]. The increased water-holding capacity of ZT soil supports soil microbial activity, organic matter decomposition, and nutrient mineralization [76]. Additionally, accelerated carbon sequestration in ZT soil helps mitigate climate change while improving soil fertility.

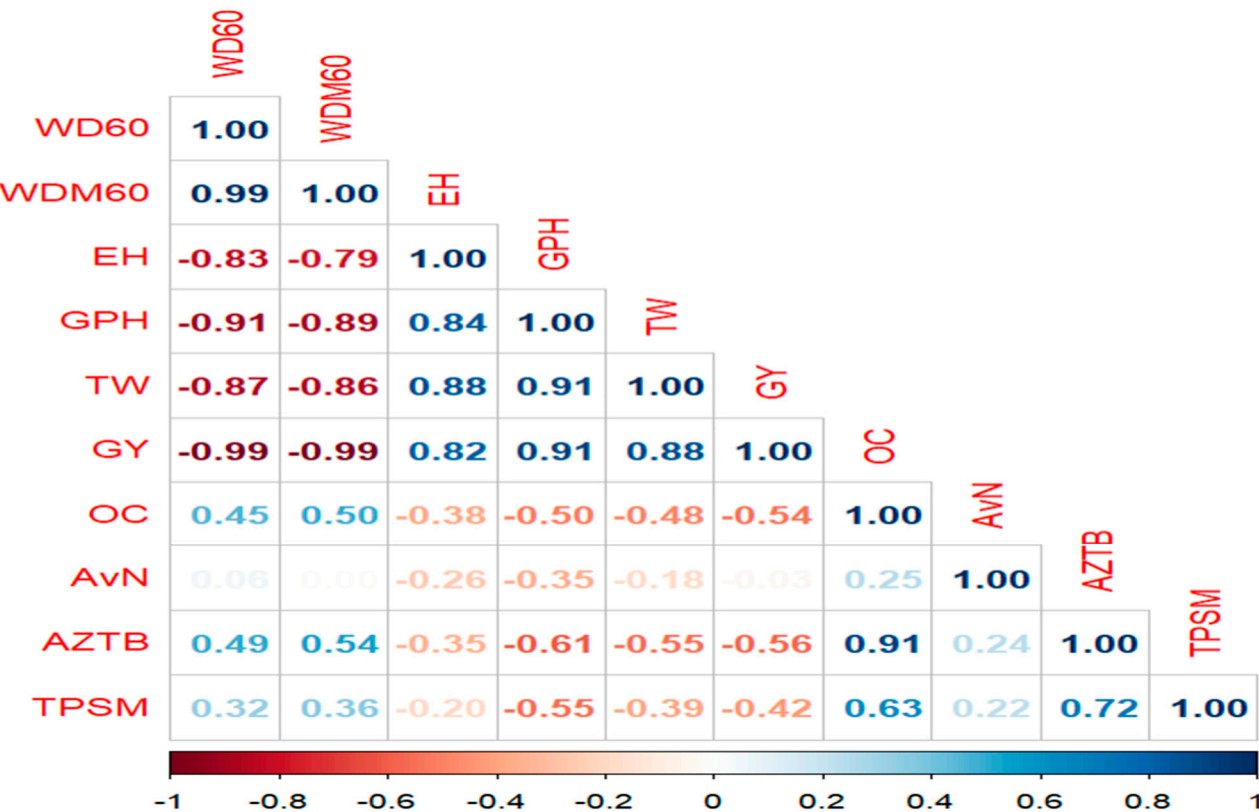

**Figure 6.** Correlation plot among different parameters of wheat cultivation across the years (mean of 3 years, WD60: weed density at 60 days after sowing (60DAS), WDM60: weed dry matter at 60DAS, EH: number of earheads, GPH: grains per earhead, TW: 1000 grain weight, GY: grain yield, OC: organic carbon in soil, AvN: available nitrogen in soil, AZTB: Azotobacter population, TPSM: total Pseudomonas population, TPSB: total phosphate solubilizing bacteria population).

Apart from the nutrient availability status of the soil, ZT promotes advanced microbial populations, including Azotobacter, Pseudomonas, and Bacillus, through a synergistic interplay of various factors involving the preservation of soil structure in $T_5$ involving ZT in all three crops in cropping sequence along with residue, creating protected microenvironments [97] and aggregates that serve as favorable niches for these microbial populations to establish and thrive [98,99]. Furthermore, ZT enables the accumulation of SOM, leading to a nutrient-rich soil environment that supports microbial growth and activity as well [100–102]. The improved soil aggregation and reduced disturbance in ZT systems enhance microbial diversity, while also safeguarding these microbes from environmental stresses [93,94]. Additionally, the enhanced water retention capacity and reduced soil erosion in ZT soil provide conducive conditions for the proliferation of Azotobacter, Pseudomonas, and Bacillus [103]. Moreover, beneficial interactions between microorganisms and plants, such as nitrogen fixation by Azotobacter and phosphate solubilization by Pseudomonas and Bacillus, are fostered in the undisturbed soil environment of ZT. Finally, the minimized chemical disturbances in ZT practices further support the growth and persistence of these microbial populations [101].

The incorporation of crop residues and the adoption of crop rotation in IWM enhance soil organic matter content [102], promoting the growth of diverse microbial communities. These beneficial microorganisms play vital roles in nutrient cycling, disease suppression, and soil health improvement, ultimately enhancing soil structure and nutrient availability [103]. Furthermore, IWM reduces the reliance on herbicides, minimizing potential negative effects on soil nutrient availability [104]. By effectively suppressing weeds, IWM allows crops to access and utilize available nutrients more efficiently [105]. The incorporation of organic matter through crop residues and green manuring practices in IWM

further enhances soil fertility and nutrient content [106], nutrient mineralization [107], and soil biological properties [97]. The combined effects of reduced weed competition, improved soil biological properties, and enhanced nutrient cycling dynamics create a favorable environment for crop growth [108].

*4.4. Economic Benefits*

The higher cost of cultivation in ZT rice–wheat systems compared to conventionally tilled systems can be attributed to several interconnected factors. Firstly, the operational cost of specialized equipment, including seed drills and precision planters, adds to the overall expenses [83]. Additionally, the reliance on high-quality hybrid or certified seeds and the need for treated seeds for successful germination further increase seed costs. ZT practices often require greater inputs of fertilizers, herbicides, and pesticides to manage weeds and pests without tillage operations, contributing to higher production costs [37,109]. The retention of crop residues on the soil surface necessitates additional investment in machinery or labor for effective residue management. Moreover, acquiring specialized knowledge and training through workshops or consultants incurs educational costs [110]. Finally, the need for risk management strategies such as crop insurance to mitigate risks associated with diseases, pests, and adverse weather conditions adds to the overall cost of cultivation [111]. While ZT offers long-term soil conservation benefits [112], careful consideration of these cost factors is essential for farmers. CT systems exhibit higher yield potential due to favorable seedbed preparation, which promotes better seed germination and reduces weed competition [91]. Effective weed control and pest management in conventionally tilled systems contribute to improved crop growth and yield, minimizing yield losses [107]. Additionally, CT offers better opportunities for pest and disease management [78], reducing the risk of damage to crops, and precise nutrient management through tillage operations enhances nutrient availability and uptake [92], leading to higher crop yields [113]. Lower input costs in CT systems, including reduced reliance on specialized equipment and inputs, contribute to higher net returns. Considering these factors collectively provides insights into the economic advantages of CT rice–wheat systems, highlighting the need for further research to bridge the yield gap and improve the economic viability of ZT [114,115]. Farmers can make informed decisions based on these considerations to optimize their production practices and maximize profitability.

Additionally, the gross returns in IWM are higher, while the net returns and B:C ratio are higher in W1 [116] due to the higher cultivation costs. With appropriate location-specific optimization of the available techniques, IWM can be recommended with regard to environmental aspects and issues such as herbicide resistance and weed shift [91,117].

## 5. Conclusions

The research findings highlight the potential of conservation tillage and integrated weed management practices to promote sustainable agriculture in the rice–wheat–green gram system. The results revealed that CT–rice along with ZT–wheat significantly reduced weed emergence, distribution, and biomass, which was ascribed to the puddling destroying the weed habitat followed by no tillage that does not allow underground weed seed to come to the surface. IWM gives superior results in controlling weed flora due to the proper combination and additive effect of weed control measures. This reduction in weed pressure positively influenced crop performance, leading to better yield-attributing characteristics for all the crops in the sequence and, in turn, increased grain and straw yield. Additionally, conservation tillage involving ZT in all crops along with residue retention significantly enhanced soil chemical and biological properties in terms of enhancement in the soil organic carbon content and nutrient availability through an augmentation in the microbial population such as *Azotobacter*, *Pseudomonas* and *Bacillus* through providing better habitat for them along with substrates through residue incorporation, thus contributing to improved soil health and fertility. However, CT–rice, followed by ZT in wheat and green gram, gives more monetary remuneration in terms of net and gross return with

a superior B:C ratio and vigorous crop growth with lesser weed pressure. Furthermore, integrated weed management practices demonstrated their ability to further suppress weed growth and enhance crop productivity. Therefore, by reducing weed pressure, improving soil chemical and biological health, and enhancing crop performance, these improved management practices offer promising avenues for farmers in the Eastern Indo-Gangetic Plain and similar agro-ecologies to achieve higher yields and profitability.

**Author Contributions:** Conceptualization, D.K.R.; methodology, D.K.R.; software, D.K.R., validation, D.K.R.; formal analysis, D.K.R.; investigation, D.K.R.; resources, D.K.R.; data curation, B.A.A., S.R., S.R.P. and S.S.; writing—original draft preparation, B.A.A., S.R., S.R.P. and S.S.; writing—review and editing, B.A.A., S.R., S.R.P, S.S., D.N. and H.G.; visualization, D.K.R. and M.F.S.; supervision, H.G. All authors have read and agreed to the published version of the manuscript.

**Funding:** The authors extend their appreciation to Prince Sattam bin Abdulaziz University for funding this research work through project number PSAU/2023/01/233562.

**Data Availability Statement:** Publicly available datasets were analyzed in this study. Some parts of the data can be found at: https://aicrp.icar.gov.in/wm/publication/annual-reports/ (accesed on 12 April 2023).

**Conflicts of Interest:** The authors declare no conflict of interest.

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
