# Peer review of "Conservation Tillage and Weed Management Influencing Weed Dynamics, Crop Performance, Soil Properties, and Profitability in a Rice–Wheat–Greengram System in the Eastern Indo-Gangetic Plain"

_agronomy, doi:10.3390/agronomy13071953_

Round 1
Reviewer 1 Report
The manuscript has been well revised, I have no further question.
Reviewer 2 Report
please see the attachment

please see the attachment
Reviewer 3 Report
Dear authors
congatulations for your work. It was a pleasure to revise such a well prepared article.
Just some points to your attention.
1) lines 39-40 should be erased
2) In all Tables I suggest adding mean values for all traits for easier comparisons by the reader.
3) Sowing densities for rice (both direct sowing 25 kg/ha and nurseries 15 kg/ha) are extremly low and outside literature rate. Moreover, yields reported are average usually harvested weith muh higher sowing densities. Please correct if misstyped or explain in text why you chose such low densities.
4) The same for greengram sowing rate.
Reviewer 4 Report
General comments: This study investigated the influence of conservation tillage and weed management effect on weed dynamics, crop performance, soil properties, and profitability in rice-wheat-greengram system in Eastern Indo-Gangetic Plains. The following comments and suggestions may be helpful to improve the manuscript.
Specific comments:
(1) Lines 39-41: “The introduction should briefly place the study in a broad context and highlight why it is important. It should define the purpose of the work and its significance. The current state”. These statements should be deleted as they are guide for authors.
(2) Section 2.2: There are too many weather details in this section, please simplify this section and keep only the descriptions that directly affected the tested results.
(3) Section 2.5: The authors stated that “To ensure representative soil samples, "V" shaped slices were created, and five random samples were collected. These samples were thoroughly mixed, and approximately 500 g of soil was taken for analysis of the parameters listed in Table 3”. Did the authors measured the parameters in Table 3 without replication since five random samples were mixed to form only one sample?
(4) Conclusions: The section is too wordy. Please simplify this section by listing several points concluded from the sections of the results and discussion.
Moderate editing of English language required.
Round 2
Reviewer 4 Report
The manuscript has been significantly improved except for the conclusion section. The conclusions should be re-written by listing several points concluded from the sections of the results and discussion, and each point cover a certain conclusion, i.e., (1)……. (2)……., please revise it.
Minor editing of English language is required.
